# HIERARCHICALLY REGULARIZED DEEP FORECASTING

## ABSTRACT

Hierarchical forecasting is a key problem in many practical multivariate forecasting applications - the goal is to simultaneously predict a large number of correlated time series that are arranged in a pre-specified aggregation hierarchy. The main challenge is to exploit the hierarchical correlations to simultaneously obtain good prediction accuracy for time series at different levels of the hierarchy. In this paper, we propose a new approach for hierarchical forecasting which consists of two components. First, decomposing the time series along a global set of basis time series and modeling hierarchical constraints using the coefficients of the basis decomposition. And second, using a linear autoregressive model with coefficients that vary with time. Unlike past methods, our approach is scalable (inference for a specific time series only needs access to its own history) while also modeling the hierarchical structure via (approximate) coherence constraints among the time series forecasts. We experiment on several public datasets and demonstrate significantly improved overall performance on forecasts at different levels of the hierarchy, compared to existing state-of-the-art hierarchical models.

## 1 INTRODUCTION

Multivariate time series forecasting is a key problem in many domains such as retail demand forecasting (Böse et al., 2017), financial predictions (Zhou et al., 2020), power grid optimization (Hyndman & Fan, 2009), road traffic modeling (Li et al., 2017), and online ads optimization (Cui et al., 2011). In many of these setting, the problem involves simultaneously forecasting a large number of possibly correlated time series for various downstream applications. In the retail domain, the time series may capture sales of items in a product inventory, and in power grids, the time series may correspond to energy consumption in a household. Often, these time series are arranged in a natural multi-level hierarchy - for example in retail forecasting, items are grouped into subcategories and categories, and arranged in a product taxonomy. In the case of power consumption forecasting, individual households are grouped into neighborhoods, counties, and cities. The hierarchical structure among the time series can usually be represented as a tree, with the leaf nodes corresponding to time series at the finest granularity, and the edges representing parent-child relationships. Figure 1 illustrates a typical hierarchy in the retail forecasting domain for time series of product sales.

In such settings, it is often required to obtain good forecasts, not just for the leaf level time-series (fine grained forecasts), but also for the aggregated time-series corresponding to higher level nodes (coarse gained forecasts). Furthermore, for interpretability and business decision making purposes, it is often desirable to obtain predictions that are roughly *coherent* or *consistent* (Hyndman et al., 2011) with respect to the hierarchy tree. This means that the predictions for each parent time-series is equal to the sum of the predictions for its children time-series. More importantly, incorporating coherence constraints in a hierarchical forecasting model captures the natural inductive bias in most hierarchical datasets, where the ground truth parent and children time series indeed adhere to additive constraints. For example, total sales of a product category is equal to the sum of sales of all items in that category.

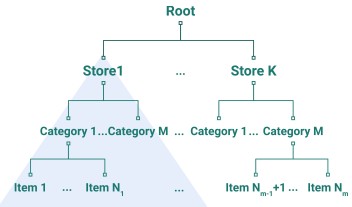

Figure 1: An example hierarchy for retail demand forecasting. The blue triangle represents the sub-tree rooted at the node *Store1* with leaves denoted by *Item i*.

Some standard approaches for hierarchical forecasting include bottom-up aggregation, or reconciliation-based approaches. Bottom-Up aggregation involves training a model to obtain predictions for the leaf nodes, and then aggregate up along the hierarchy tree to obtain predictions for higher-level nodes. Reconciliation methods (Ben Taieb & Koo, 2019; Taieb et al., 2017; Van Erven & Cugliari, 2015; Hyndman et al., 2016; Wickramasuriya et al., 2015; 2020; Panagiotelis et al., 2020) make use of a trained model to obtain predictions for all nodes of the tree, and then, in a separate post-processing phase, *reconcile* (or modify) them using various optimization formulations to obtain coherent predictions. Both of these approaches suffer from shortcomings in term of either aggregating noise as one moves to higher level predictions (bottom-up aggregation), or not jointly optimizing the forecasting predictions along with the coherence constraints (for instance, reconciliation).

At the same time, there have been several recent advances on using Deep Neural Network models for multivariate forecasting, including Recurrent Neural Network (RNN), Convolutional Neural Network (CNN) architectures (Salinas et al., 2020; Oreshkin et al., 2019; Rangapuram et al., 2018; Benidis et al., 2020), and models designed for multivariate time series based on dimensionality reduction techniques (Sen et al., 2019; Wang et al., 2019; Nguyen & Quanz, 2021; Salinas et al., 2019; Rasul et al., 2020; de Bézenac et al., 2020), that have been shown to outperform classical time-series models such as autoregressive and exponential smoothing models (McKenzie, 1984; Hyndman et al., 2008; Hyndman & Athanasopoulos, 2018), especially for large datasets. However, most of these approaches do not explicitly address the question of how to model the hierarchical relationships in the dataset. Deep forecasting models based on Graph Neural Networks (GNN) (Bai et al., 2020; Cao et al., 2020; Yu et al., 2017; Li et al., 2017; Wu et al., 2020) do offer a general framework for learning on graph-structured data. However it is well known (Bojchevski et al., 2020) that GNNs are hard to scale for learning on graphs with a very large number of nodes, which in real-world settings such as retail forecasting, could involve hundreds of thousands of time series. More importantly, a desirable practical feature for multi-variate forecasting models is to let the prediction of future values for a particular time series only require as input historical data from that time series (along with covariates), without requiring access to historical data from all other time series in the hierarchy. This allows for scalable training and inference of such models using mini-batch gradient descent, without requiring each batch to contain all the time series in the hierarchy. This is often not possible for GNN-based forecasting models, which require batch sizes of the order of the number of time series.

**Problem Statement:** Based on the above motivations, our goal is to design a hierarchical forecasting model with the following requirements: 1) The model can be trained using a single-stage pipeline on all the time series data, without any separate post-processing, 2) The model captures the additive coherence constraints along the edges of the hierarchy, 3) The model is efficiently trainable on large datasets, without requiring, for instance, batch sizes that scale with the number of time series.

We propose a principled methodology to address all these above requirements for hierarchical forecasting. Our model comprises of two components, both of which can support coherence constraints. The first component is purely a function of the historical values of a time series, without distinguishing between the individual time series themselves in any other way. Coherence constraints on such a model correspond to imposing an additivity property on the prediction function - which constrains the model to be a linear autoregressive model. However, crucially, our model uses time-varying autoregressive coefficients that can themselves be nonlinear functions of the timestamp and other global features (linear versions of time-varying AR have been historically used to deal with non-stationary signals (Sharman & Friedlander, 1984)). We will refer to this component as the *time-varying autoregressive model*.

The second component focuses on modeling the global temporal patterns in the dataset through identifying a small set of temporal *global basis functions*. The basis time-series, when combined in different ways, can express the individual dynamics of each time series. In our model, the basis time-series are encoded in a trained seq-2-seq model (Sutskever et al., 2014) model in a functional form. Each time series is then associated with a learned embedding vector that specifies the weights for decomposition along these basis functions. Predicting a time series into the future using this model then just involves extrapolating the global basis functions and combining them using its weight vector, without explicitly using the past values of that time series. The coherence constraints therefore only impose constraints on the embedding vectors of each time series, which can be easily modeled by a hierarchical regularization function. We call this component a *basis decomposition model*. As we will see, this part of the model is only approximately coherent unless the embedding constraints hold exactly. In particular, in this paper, we focus on improving model accuracy rather than preserving

exact coherency. In Section A.2, we also provide theoretical justification for how such hierarchical regularization using basis decomposition results in improved prediction accuracy.

We experimentally evaluate our model on multiple publicly available hierarchical forecasting datasets. We compare our approach to state-of-the-art (non-hierarchical) deep forecasting models, GNN-based models and reconciliation models, and show that our approach can obtain consistently more accurate predictions at all levels of the hierarchy tree.

## 2 RELATED WORK ON DEEP HIERARCHICAL MODELS

In addition to the works referenced in the previous section, we now discuss a few papers that are more relevant to our approach. Specifically, we discuss some recent deep hierarchical forecasting methods that do not require a post-processing reconciliation step. Hierarchical forecasting methods can be roughly divided into two categories: *point forecasters* and *probabilistic forecasters*. Mishchenko et al. (2019) propose a point-forecasting approach which imposes coherency on a base model via $\ell_2$ regularization on the predictions. Gleason (2020) extend the idea further to impose the hierarchy on an embedding space rather than the predictions directly. SHARQ (Han et al., 2021) follows a similar $\ell_2$ regularization based approach as Mishchenko et al. (2019), and also extends the idea to probabilistic forecasting. Their model is trained separately for each of the hierarchical levels starting from the leaf level, thus requiring a separate prediction model for each level.

Probabilistic forecasting methods include, Hier-E2E (Rangapuram et al., 2021) which produces perfectly coherent forecasts by using a projection operation on base predictions from a DeepVAR model (Salinas et al., 2019). It requires the whole hierarchy of time series to be fed as input to the model leading to a large number of parameters, and hence does not scale well to large hierarchies. Yanchenko et al. (2021) take a fully Bayesian approach by modelling the hierarchy using conditional distributions.

## 3 PROBLEM SETTING

We are given a set of $N$ coherent time series of length $T$, arranged in a pre-defined hierarchy consisting of $N$ nodes. At time step $t$, the time series data can be represented as a vector $\boldsymbol{y}_t \in \mathbb{R}^N$ denoting the time series values of all $N$ nodes. We compactly denote the set of time series for all $T$ steps as a matrix $\boldsymbol{Y} = [\boldsymbol{y}_1, \cdots, \boldsymbol{y}_T]^\top \in \mathbb{R}^{T \times N}$. Also define $\boldsymbol{y}^{(i)}$ as the $i$th column of the matrix $\boldsymbol{Y}$ denoting all time steps of the $i$ th time series, and $\boldsymbol{y}_t^{(i)}$ as the $t$ th value of the $i$ th time series. We compactly denote the $H$-step history of $\boldsymbol{Y}$ by $\boldsymbol{Y}_{\mathcal{H}} = [\boldsymbol{y}_{t-H}, \cdots, \boldsymbol{y}_{t-1}]^\top \in \mathbb{R}^{H \times N}$ and the $H$-step history of $\boldsymbol{y}^{(i)}$ by $\boldsymbol{y}_{\mathcal{H}}^{(i)} = [\boldsymbol{y}_{t-H}^{(i)}, \cdots, \boldsymbol{y}_{t-1}^{(i)}] \in \mathbb{R}^H$. Similarly define the $F$-step future of $\boldsymbol{Y}$ as $\boldsymbol{Y}_{\mathcal{F}} = [\boldsymbol{y}_t, \cdots, \boldsymbol{y}_{t+F-1}]^\top \in \mathbb{R}^{F \times N}$. We use the $\widehat{\cdot}$ notation to denote predicted values, for example $\widehat{\boldsymbol{Y}}_{\mathcal{F}}$, $\widehat{\boldsymbol{y}}_{\mathcal{F}}$ and $\widehat{\boldsymbol{y}}_t$.

Time series forecasts can often be improved by using features as input to the model along with historical time series. The features often evolve with time, for example, categorical features such as *type of holiday*, or continuous features such as *time of the day*. We denote the matrix of such features by $\boldsymbol{X} \in \mathbb{R}^{T \times D}$, where the $t$ th row denotes the $D$-dimensional feature vector at the $t$ time step. For simplicity, we assume that the features are *global*, meaning that they are shared across all time series. We similarly define $\boldsymbol{X}_{\mathcal{H}}$ and $\boldsymbol{X}_{\mathcal{F}}$ as above.

**Hierarchically Coherent Time Series.** We assume that the time series data are coherent, that is, they satisfy the *sum constraints* over the hierarchy. The time series at each node of the hierarchy is the equal to the sum of the time series of its children, or equivalently, equal to the sum of the leaf time series of the sub-tree rooted at that node. Figure 1 shows an example of a sub-tree rooted at a node.

As a result of aggregation, the data can have widely varying scales with the values at higher level nodes being magnitudes higher than the leaf level nodes. It is well known that neural network training is more efficient if the data are similarly scaled. Hence, in this paper, we work with rescaled time series data. The time series at each node is downscaled by the number of leaves in the sub-tree rooted at the node, so that now they satisfy *mean constraints* rather than sum constraints described above. Denote by $\mathcal{L}(p)$, the set of leaf nodes of the sub-tree rooted at $p$. Hierarchically coherent data satisfy

the following *data mean property*,

$$\boldsymbol{y}^{(p)} = \frac{1}{|\mathcal{L}(p)|} \sum_{i \in \mathcal{L}(p)} \boldsymbol{y}^{(i)} \quad \textit{(Data Mean Property).} \tag{1}$$

## 4 HIERARCHICALLY REGULARIZED DEEP FORECASTING - HIRED

We now introduce the two components in our model, namely the *time-varying AR model* and the *basis decomposition model*. As mentioned in the introduction a combination of these two components satisfy the three requirements in our problem statement. In particular, we shall see that the coherence property plays a central role in both the components. For simplicity, in this section, all our equations will be for forecasting one step into the future ($F = 1$), even though all the ideas trivially extend to multi-step forecasting. The defining equation of our model can be written as,

$$\begin{aligned}
\widehat{\boldsymbol{y}}_{\mathcal{F}}^{(i)} &= f(\boldsymbol{y}_{\mathcal{H}}^{(i)}, \boldsymbol{X}_{\mathcal{H}}, \boldsymbol{X}_{\mathcal{F}}, \boldsymbol{Z}_{\mathcal{H}}, \theta_i) \\
&= \underbrace{\left\langle \boldsymbol{y}_{\mathcal{H}}^{(i)}, a(\boldsymbol{X}_{\mathcal{H}}, \boldsymbol{X}_{\mathcal{F}}, \boldsymbol{Z}_{\mathcal{H}}) \right\rangle}_{\text{Time varying AR (TVAR)}} + \underbrace{\left\langle \theta_i, b(\boldsymbol{X}_{\mathcal{H}}, \boldsymbol{X}_{\mathcal{F}}, \boldsymbol{Z}_{\mathcal{H}}) \right\rangle}_{\text{Basis decomposition (BD)}}.
\end{aligned} \tag{2}$$

In the above equation, $\boldsymbol{Z}_{\mathcal{H}}$ is a latent state vector that contains some summary temporal information about the whole dataset, and $\theta_i$ is the embedding/weight vector for time-series $i$ in the basis decomposition model. $\boldsymbol{Z}_{\mathcal{H}}$ can be a relatively low-dimensional temporally evolving variable that represents some information about the global state of the dataset at a particular time. We use the *Non-Negative Matrix Factorization* (NMF) algorithm by Gillis & Vavasis (2013) to select a small set of representative time series that encode the global state. If the indices of the selected representative time-series is denoted by $\{i_1, \cdots, i_R\}$ ($R$ denotes the *rank* of the factorization), then we define $\boldsymbol{Z} = [\boldsymbol{Y}^{(i_1)}, \cdots, \boldsymbol{Y}^{(i_R)}] \in \mathbb{R}^{T \times R}$. Note that we only feed the past values $\boldsymbol{Z}_{\mathcal{H}}$ as input to the model, since future values are not available during forecasting. Also, note that the final basis time-series can be a non-linear function of $\boldsymbol{Z}_{\mathcal{H}}$. In our experiments, $R$ is tuned but is always much much less than $N$. $a$ and $b$ are functions not dependent on $\boldsymbol{y}_{\mathcal{H}}^{(i)}$ and $\theta_i$. We will provide more details as we delve into individual components.

**Time-Varying AR (TVAR):** The first part of the expression in equation 2 denoted by *Time Varying AR (TVAR)* resembles a linear auto-regressive model with coefficients $a(\boldsymbol{X}_{\mathcal{H}}, \boldsymbol{X}_{\mathcal{F}}, \boldsymbol{Z}_{\mathcal{H}}) \in \mathbb{R}^H$, that are a function of the input features, and thus can change with time. The AR parameters of this model are shared across all time series and hence do not encode any time-series specific information, a drawback that is overcome by the Basis Decomposition part of our model. This component is *coherent by design* because it is a shared linear AR model. However, even though the AR weights are shared across all the time-series at a given time-point, they can crucially change with time, thus lending more flexibility to the model.

*Implementation:* In order to model the sequential nature of the data, we use an LSTM encoder to encode the past $\boldsymbol{X}_{\mathcal{H}}$ and $\boldsymbol{Z}_{\mathcal{H}}$. Then, we use a fully connected (FC) decoder for predicting the auto-regressive weights. Similar to Wen et al. (2017)'s multi-horizon approach, we use a different head of the decoder for each future time step resulting in a $F$-headed decoder producing $F$-step predictions for TVAR weights. The decoder also takes as input the future covariates $\boldsymbol{X}_{\mathcal{F}}$ if available. The produced weights are then multiplied (inner product) to the history to produce the final TVAR predictions. We illustrate this architecture in Figure 2 (right).

**Basis Decomposition (BD) with Hierarchical Regularization:** Now we come to the second part of our model in equation 2. As discussed before, this part of the model has per time-series adaptivity, as different time-series can have different embeddings. It resembles an expansion of the time series on a set of basis functions $b(\boldsymbol{X}_{\mathcal{H}}, \boldsymbol{X}_{\mathcal{F}}, \boldsymbol{Z}_{\mathcal{H}}) \in \mathbb{R}^K$, with the basis weights/embedding for time series $i$ denoted by $\theta_i \in \mathbb{R}^K$. Both the basis functions and the time series specific weights are learned from the data, rather than fixing a specific form such as Fourier or Wavelet basis.

The idea of using a basis has also been recently invoked in the time-series literature (Sen et al., 2019; Wang et al., 2019). The basis recovered in the implementation of Wang et al. (2019) is allowed to vary for each individual time-series and therefore is not a true basis. Sen et al. (2019) do explicitly recover an approximate basis in the training set through low-rank matrix factorization regularized

by a deep global predictive model alternatingly trained on the basis vectors, thus not amenable to end-to-end optimization. We shall see that our model can be trained in an end-to-end manner.

*Embedding Regularization for Approximate Coherency:* The TVAR part of our model is coherent by design due to its linearity. The BD model however requires the embeddings of the time-series to satisfy the mean property along the hierarchy. This directly translates to coherency of the predictions due to linearity with respect to $\theta$.

$$\theta_p = \frac{1}{|\mathcal{L}(p)|} \sum_{i \in \mathcal{L}(p)} \theta_i \quad \textit{(Embedding Mean Property)}, \tag{3}$$

We impose this constraint approximately via an $\ell_2$ regularization on the embedding.

$$E_{\text{reg}}(\boldsymbol{\theta}) = \sum_{p=1}^{N} \sum_{i \in \mathcal{L}(p)} \|\theta_p - \theta_i\|_2^2. \tag{4}$$

The purpose of this regularizer is two fold. Firstly, we observe that, when the leaf embeddings are kept fixed, the regularizer is minimized when the embeddings satisfy the mean property (3), thus encouraging coherency in the predictions. Secondly, it also encodes the inductive bias present in the data corresponding to the hierarchical additive constraints. We provide some theoretical justification for this hierarchical regularization in Section 5.

*Implementation:* As before, we use an LSTM encoder to encode the past $\boldsymbol{X}_{\mathcal{H}}$ and $\boldsymbol{Z}_{\mathcal{H}}$. Then, we use the encoding from the encoder along with the future features $\boldsymbol{X}_{\mathcal{F}}$ (sequential in nature) and pass them through an LSTM decoder to yield the $F$-step basis predictions which are then multiplied with the embedding (inner product) to produce the final BD predictions. We illustrate this architecture in the top right of Figure 2. Thus, a functional representation of the basis time-series is implicitly maintained within the trained weights of the basis generating seq-2-seq model. Note that the embeddings are also trained in our end-to-end model. We illustrate this architecture in Figure 2 (left).

We emphasize that the main ideas in our model are agnostic to the specific type of neural network architecture used. For our experiments, we specifically use an LSTM architecture (Hochreiter & Schmidhuber, 1997) for the encoder and decoder. Other types of architectures including transformers (Vaswani et al., 2017) and temporal convolution networks (Borovykh et al., 2017) can also be used.

**Loss Function:** During training, we minimize the mean absolute error (MAE) of the predictions along with the embedding regularization term introduced above (our method generalizes to other losses too, such as mean square error, or mean absolute percentage error). For regularization weight $\lambda_E$, and $\widehat{\boldsymbol{y}}_{\mathcal{F}}^{(i)}$ defined as Eq (2), and $\Theta$ denoting the trainable parameters of $a, b$, our training loss function is,

$$\ell(\Theta, \boldsymbol{\theta}) = \underbrace{\sum_i \sum_{\mathcal{F}} |\boldsymbol{y}_{\mathcal{F}}^{(i)} - \widehat{\boldsymbol{y}}_{\mathcal{F}}^{(i)}|}_{\text{Prediction loss}} + \underbrace{\lambda_E E_{\text{reg}}(\boldsymbol{\theta})}_{\text{Embedding regularization}}. \tag{5}$$

Note that the time-series dependent part of the loss function can be easily mini-batched and the embeddings are not memory intensive.

## 5 THEORETICAL JUSTIFICATION FOR HIERARCHICAL MODELING

In this section, we theoretically analyze the benefits of modeling hierarchical constraints in a much simplified setting, and show how it can result in provably improved accuracy, under some assumptions. Since analyzing our actual deep non-linear model for an arbitrary hierarchical set of time series can be complex, we make some simplifying assumptions to the problem and model. We assume that all the time series in the dataset is a linear combination of a small set of basis time series. That is, $\boldsymbol{Y} = \boldsymbol{B}\boldsymbol{\theta} + \boldsymbol{w}$, where $\boldsymbol{B} \in \mathbb{R}^{T \times K}$ denotes the set of basis vectors, $\boldsymbol{\theta} = [\theta_1, \cdots, \theta_N] \in \mathbb{R}^{K \times N}$ denotes the set of weight vectors used in the linear combination for each time series, and $\boldsymbol{w} \in \mathbb{R}^{T \times N}$ denotes the noise matrix sampled i.i.d as $w \sim \mathcal{N}(0, \sigma^2)$ for the leaf nodes. A classical example of such a basis set can be a small subset of Fourier or Wavelet basis (Strang, 1993; van den Oord et al., 2016) that is relevant to the dataset. Note that we ignore the TVAR model for the sake of analysis and focus mainly on the BD model which includes the hierarchical regularization.

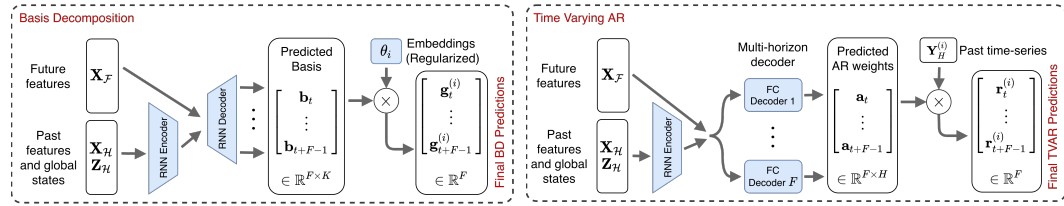

Figure 2: In this figure we show the architectures of our two model components separately. On the left we show the BD model, where the seq-2-seq model implicitly maintains the basis in a functional form. Note that the time-series specific weights $\{\theta_i\}$ are also trained. On the right, we show the TVAR model. The fully connected decoder has a different prediction head for each future time-point.

---

**Algorithm 1:** Basis Recovery

**Input:** Observed $y$, basis dict $\bar{B}$, regularization parameter $\lambda_L$
**Output:** Estimated basis $B$
$\widehat{\alpha}_0 \leftarrow$
$\quad \underset{\alpha \in \mathbb{R}^n}{\operatorname{argmin}} \frac{1}{2T} \|y_0 - \bar{B}\alpha\|_2^2 + \lambda_L \|\alpha\|_1$
Estimate support $\widehat{S} = \{i \mid |\widehat{\alpha}_0| > 0\}$
Estimate true basis $B \leftarrow \bar{B}_{\widehat{S}}$

---

**Algorithm 2:** Parameter Recovery

**Input:** Observed time series $y$, estimated basis $B$, regularization parameter $\lambda_E$
**Output:** Estimated parameters $\theta$
$\widehat{\theta}_0 \leftarrow \operatorname{argmin}_{\theta_0} \frac{1}{T} \|y_0 - B\theta_0\|_2^2$
**for** $n \in \mathcal{L}(0)$ **do**
$\quad \Big| \ \widehat{\theta}_n \leftarrow \operatorname{argmin}_{\theta_n} \frac{1}{T} \|y_n - B\theta_n\|_2^2 + \lambda_E \|\widehat{\theta}_0 - \theta_n\|_2^2.$
**end**

---

In this section, we consider a 2-level hierarchy of time-series, consisting of a single root node (indexed by 0) with $L$ children (denoted by $\mathcal{L}(0)$). We will also assume that instead of learning the $K$ basis vectors $B$ from scratch, the $K$ basis vectors are assumed to come from a much larger dictionary $\bar{B} \in \mathbb{R}^{T \times D}$ of $D \ (\gg K)$ vectors that is fixed and known to the model. While the original problem learns the basis and the coefficients $\theta$ simultaneously, in this case the goal is to select the basis from among a larger dictionary, and learn the coefficients $\theta$.

We analyze this problem, and show that under the reasonable assumption of the parent embedding $\theta_0$ being close to all the children embeddings $\theta_n$, using the hierarchical constraints can result in a mean-square error at the leaf nodes that is a multiplicative factor $L$ smaller than the optimal mean-square error of any model that does not use the hierarchical constraints. Our proposed HiRED model, when applied in this setting would result in the following (hierarchically) regularized regression problem:

$$\min_{\boldsymbol{\theta}} \frac{1}{NT} \|\boldsymbol{y} - \boldsymbol{B}\boldsymbol{\theta}\|_2^2 + \lambda \sum_{n \in \mathcal{L}(0)} \|\theta_0 - \theta_n\|_2^2. \tag{6}$$

For the sake of analysis, we instead consider a two-stage version, described in Algorithm 1 and Algorithm 2: we first recover the support of the basis using Basis Pursuit (Chen et al., 2001). We then estimate the parameters of the root node, which is then plugged-in to solve for the parameters of the children node. We also define the baseline (unregularized) optimization problem for the leaf nodes that does not use any hierarchical information, as

$$\tilde{\theta}_n = \underset{\theta_n}{\operatorname{argmin}} \frac{1}{T} \|y_n - \boldsymbol{B}\theta_n\|_2^2 \quad \forall n \in \mathcal{L}(0). \tag{7}$$

The basis support recovery follows from standard analysis (Wainwright, 2009) detailed in Lemma 1 in the Appendix. We focus on the performance of Algorithm 2 here. The following theorem bounds the error of the unregularized ($\tilde{\theta}_n$) and the hierarchically-regularized ($\widehat{\theta}_n$, see Algorithm 2) optimization solutions. A proof of the theorem can be found in Appendix A.2.

**Theorem 1.** *Suppose the rows of $\boldsymbol{B}$ are norm bounded as $\|\boldsymbol{B}_i\|_2 \leq r$, and $\|\theta_n - \theta_0\|_2 \leq \beta$. Define $\Sigma = \boldsymbol{B}^T\boldsymbol{B}/T$ as the empirical covariance matrix. For $\lambda_E = \frac{\sigma^2 K}{T \beta^2}$, $\tilde{\theta}_n$ and $\widehat{\theta}_n$ can be bounded as,*

$$\mathbb{E}\|\tilde{\theta}_n - \theta_n\|_\Sigma^2 \leq \frac{\sigma^2 K}{T}, \quad \mathbb{E}\|\widehat{\theta}_n - \theta_n\|_\Sigma^2 \leq 3\frac{\sigma^2 K}{T} \frac{1}{1 + \frac{\sigma^2 K}{T r^2 \beta^2}} + 6\frac{\sigma^2 K}{TL}. \tag{8}$$

In fixed design linear regression $\|\widehat{\theta}_n - \theta_n\|_{\Sigma}^2 = \|\boldsymbol{B}(\widehat{\theta}_n - \theta_n)\|^2$ is the population squared error (see Appendix A.5 for a bound on the parameter estimation error). The gains due to the regularization can be understood by considering the case when $\beta$ is upper bounded by a sufficiently small quantity. Note that an upper bound on $\beta$ essentially implies that the children time-series have structural similarities as further elaborated in Appendix A.5. We show that the above assumption yields a smaller upper bound on the error. In fact, if $\beta = o(\sqrt{K/T})$, then the numerator $1 + \frac{\sigma^2 K}{Tr^2\beta^2}$ in Eq. (8) is $\omega(1)$ resulting in $\mathbb{E}\|\widehat{\theta}_n - \theta_n\|_{\Sigma}^2 = o(\frac{\sigma^2 K}{T})$ which decays faster than $\frac{\sigma^2 K}{T}$. Furthermore, if $\beta$ is even smaller as $\beta = O(\sqrt{K/LT})$, then following similar calculations, $\mathbb{E}\|\widehat{\theta}_n - \theta_n\|_{\Sigma}^2 = O(\frac{\sigma^2 K}{LT})$ which is again smaller than the unregularized bound.

## 6 EXPERIMENTS

We implemented our proposed model in Tensorflow (Abadi et al., 2016) and compared against multiple baselines on popular hierarchical time-series datasets.

**Datasets.** We experimented with three hierarchical forecasting datasets - Two retail forecasting datasets, M5 (M5, 2020) and Favorita (Favorita, 2017); and the Tourism (Tourism, 2019) dataset consisting of tourist count data. The history length and forecast horizon $(H, F)$ were set to (28, 7), (28, 7) and (24, 4), for Favorita, M5 and Tourism respectively. More information can be found in Appendix B.2. We divide each of the datasets into training, validation and test sets, with details on the splits provided in Appendix B.3.

**Baselines.** We compare our proposed approach HIRED with the following baseline models: (i) *RNN* - we use a seq-2-seq model shared across all the time series, (ii) *DeepGLO* (Sen et al., 2019), (iii) *DCRNN* (Li et al., 2017), a GNN based approach where we feed the hierarchy tree as the input graph, (iv) *Deep Factors (DF)* (Wang et al., 2019), (v) $L_2Emb$ (Gleason, 2020), which is an improvement over Mishchenko et al. (2019), (vi) *SHARQ* (Han et al., 2021), (vii) *HierE2E* (Rangapuram et al., 2021) - the standard implementation produces probabilistic forecasts, we however adapt it to point forecasts by using their projection step on top of a seq-2-seq model. We use code publicly released by the authors for DeepGLO[1] and DCRNN[2]. We implemented our own version of DeepFactors, HierE2E, and $L_2$Emb for a fair comparison, since the official implementations either make rolling probabilistic forecasts, or use a different set of covariates. Additionally, we also compare with the recent *ERM* (Ben Taieb & Koo, 2019) reconciliation method applied to the base forecasts from the RNN model, denoted as (vii) *RNN+ERM*. It has been shown in (Ben Taieb & Koo, 2019) to outperform many previous reconciliation techniques such as MinT (Wickramasuriya et al., 2019). For a fair comparison, we use the same Mean Absolute Error (MAE) loss for all the compared methods, and make sure that all the models have approximately the same number of parameters. Further details about the baselines and training parameters can be found in Appendix B.

**Metrics.** We compare the accuracy of the various approaches with respect to two metrics: (i) *weighted absolute percentage error (WAPE)*, and (ii) *symmetric mean absolute percentage error (SMAPE)*. A description of these metrics can be found in Appendix B.1. We report the metrics on the test data, for each level of the hierarchy (with level 0 denoting the root) in Table 1. As a measure of the aggregate performance across all the levels, we also report the mean of the metrics in all the levels of the hierarchy denoted by *Mean*.

### 6.1 RESULTS

Table 1 shows the averaged test metrics for M5, Favorita, and Tourism datasets. We present only the *Mean* metrics for the Tourism dataset due to lack of space. Complete results with confidence intervals for all the three datasets can be be found in Appendix B.6.

We find that for all three datasets, our proposed model either yields the smallest error or close to the smallest error across most metrics and most levels. In particular, we find that our proposed method achieves the smallest errors in the mean column for all datasets in terms of WAPE and SMAPE, thus indicating good performance generally across all levels. We find that RNN+ERM in general, yields

---

[1] https://github.com/rajatsen91/deepglo
[2] https://github.com/liyaguang/DCRNN/

Table 1: The tables show the WAPE/SMAPE test metrics for the M5, Tourism, and Favorita datasets, averaged over 10 runs. We present only the Mean metrics for the Tourism dataset due to lack of space. A complete set of results including the standard deviations can be found in Appendix B.6. In-coherent and coherent baselines are separated by a horizontal line.

| M5 | Level 0 | Level 1 | Level 2 | Level 3 | Mean |
|---|---|---|---|---|---|
| HiReD | **0.048 / 0.048** | **0.055 / 0.053** | **0.072 / 0.077** | 0.279 / 0.511 | **0.113 / 0.172** |
| RNN | 0.059 / 0.059 | 0.083 / 0.083 | 0.085 / 0.098 | 0.282 / 0.517 | 0.127 / 0.189 |
| DF | 0.055 / 0.056 | 0.061 / 0.060 | 0.076 / 0.085 | 0.272 / **0.501** | 0.116 / 0.176 |
| DeepGLO | 0.077 / 0.081 | 0.087 / 0.092 | 0.099 / 0.113 | 0.278 / 0.538 | 0.135 / 0.206 |
| DCRNN | 0.078 / 0.079 | 0.096 / 0.092 | 0.165 / 0.193 | 0.282 / 0.512 | 0.156 / 0.219 |
| $L_2$Emb | 0.055 / 0.056 | 0.064 / 0.063 | 0.080 / 0.092 | **0.269 / 0.501** | 0.117 / 0.178 |
| SHARQ | 0.093 / 0.096 | 0.071 / 0.062 | 0.099 / 0.094 | 0.277 / 0.528 | 0.135 / 0.195 |
| RNN+ERM | 0.052 / 0.052 | 0.066 / 0.071 | 0.084 / 0.104 | 0.286 / 0.520 | 0.122 / 0.187 |
| Hier-E2E | 0.152 / 0.160 | 0.152 / 0.158 | 0.152 / 0.181 | 0.396 / 0.615 | 0.213 / 0.278 |

| Tourism | Mean |
|---|---|
| HiReD | **0.186 / 0.322** |
| RNN | 0.211 / 0.333 |
| DF | 0.204 / 0.334 |
| DeepGLO | 0.199 / 0.346 |
| DCRNN | 0.281 / 0.392 |
| $L_2$Emb | 0.215 / 0.342 |
| SHARQ | 0.229 / 0.378 |
| RNN+ERM | 0.251 / 0.417 |
| Hier-E2E | 0.208 / 0.340 |

| Favorita | Level 0 | Level 1 | Level 2 | Level 3 | Mean |
|---|---|---|---|---|---|
| HiReD | 0.061 / **0.061** | **0.094 / 0.182** | **0.127 / 0.267** | 0.210 / **0.322** | **0.123 / 0.208** |
| RNN | 0.067 / 0.068 | 0.114 / 0.197 | 0.134 / 0.290 | **0.203** / 0.339 | 0.130 / 0.223 |
| DF | 0.064 / 0.064 | 0.110 / 0.194 | 0.135 / 0.291 | 0.213 / 0.343 | 0.130 / 0.223 |
| DeepGLO | 0.098 / 0.088 | 0.126 / 0.197 | 0.156 / 0.338 | 0.226 / 0.404 | 0.151 / 0.256 |
| DCRNN | 0.080 / 0.080 | 0.120 / 0.212 | 0.134 / 0.328 | **0.204** / 0.389 | 0.134 / 0.252 |
| $L_2$Emb | 0.070 / 0.070 | 0.114 / 0.199 | 0.136 / 0.276 | 0.207 / **0.321** | 0.132 / 0.216 |
| SHARQ | 0.088 / 0.085 | 0.142 / 0.199 | 0.156 / 0.335 | 0.230 / 0.404 | 0.154 / 0.256 |
| RNN+ERM | **0.056 / 0.058** | 0.103 / 0.185 | 0.129 / 0.283 | 0.220 / 0.348 | 0.127 / 0.219 |
| Hier-E2E | 0.120 / 0.125 | 0.206 / 0.334 | 0.247 / 0.448 | 0.409 / 0.573 | 0.245 / 0.370 |

an improvement over the base RNN predictions for the higher levels closer to the root node (Levels 0 and 1), while, worsening at the lower levels. DCRNN, despite using the hierarchy as a graph also does not perform as well as our approach, especially in Tourism and M5 Datasets - possibly due to the fact that a GNN is not the most effective way to model the tree hierarchies. We notice that HierE2E performs reasonably well for the smaller Tourism while performing badly for the larger M5 and Favorita datasets - a possible explanation being that this is a VAR model that requires much more parameters to scale to large datasets. Therefore, for HierE2E we perform experiments with $50\times$ more parameters for M5 and Favorita and report the results in Table 4 in the appendix, showing that while the results improve, it still performs much worse than our model. Overall, we find that our proposed method consistently works better or at par with the other baselines at all hierarchical levels.

**Ablation study.** Next, we perform an ablation study of our proposed model to understand the effects of its various components, the results of which are presented in Table 2. We compare our proposed model, to the same model without any regularization (set $\lambda_E = 0$ in Eq (5)), and a model consisting of only TVAR. We find that both these components in our model are important, and result in improved accuracy in most metrics.

Table 2: We report the test WAPE/SMAPE metrics for an ablation study on the M5 dataset, for each of the components in the HiReD model. We compare our model with two ablated variants: first, we remove the regularization ($\lambda_E = 0$), and second, we remove the BD component (TVAR only).

| M5 Abl | Level 0 | Level 1 | Level 2 | Level 3 | Mean |
|---|---|---|---|---|---|
| HiReD | **0.048 / 0.048** | **0.055 / 0.053** | **0.072 / 0.077** | **0.279 / 0.511** | **0.113 / 0.172** |
| $\lambda_E = 0$ | 0.054 / 0.054 | 0.058 / 0.056 | 0.074 / 0.078 | **0.279** / 0.513 | 0.116 / 0.175 |
| TVAR only | 0.050 / 0.049 | 0.064 / 0.065 | 0.084 / 0.086 | 0.288 / 0.520 | 0.122 / 0.180 |

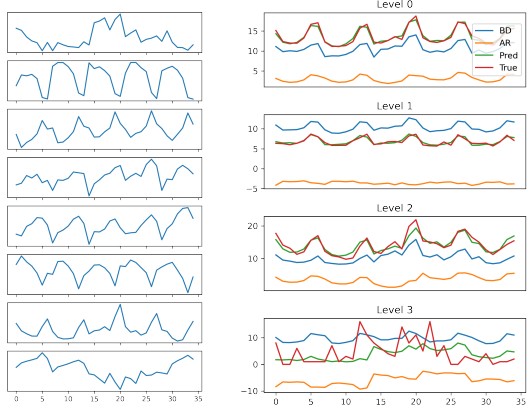

| | | L0 | L1 | L2 | L3 |
|---|---|---|---|---|---|
| Favorita | HiReD | **0.004** | **0.004** | **0.003** | - |
| | $\lambda_E = 0$ | 0.012 | 0.013 | 0.010 | - |
| | RNN | 0.043 | 0.044 | 0.042 | - |
| M5 | HiReD | **0.030** | **0.034** | **0.034** | - |
| | $\lambda_E = 0$ | 0.035 | 0.040 | 0.039 | - |
| | RNN | 0.042 | 0.057 | 0.047 | - |
| Tourism | HiReD | 0.092 | **0.079** | **0.066** | 0.060 |
| | $\lambda_E = 0$ | **0.085** | 0.082 | 0.067 | **0.059** |
| | RNN | 0.097 | 0.089 | 0.082 | 0.083 |

Figure 4: Coherency metric for all our datasets, at all hierarchical levels. Leaf node metrics are identically zero, and hence not reported in the table. Leaf nodes for Favorita and M5 are denoted by L3. Tourism has 5 hierarchical levels and hence L3 values are reported in this case.

Figure 3: Left: Plots of the basis generated on the validation set of the M5 dataset over 35 days. Right: We plot the true time series over the same time period, and compare it with the predicted time series, AR predictions and BD predictions.

**Coherence.** We also compare the coherence of our predictions to that of the RNN model and an ablated model with $\lambda_E = 0$. Specifically, for each node $p$ we measure the deviation of our forecast from $c^{(p)} = 1/\mathcal{L}(p) \sum_{i \in \mathcal{L}(p)} \widehat{y}^{(i)}$, the mean of the leaf node predictions of the corresponding sub-tree. Perfectly coherent predictions will have a zero deviation from this quantity. In Table 4, we report the WAPE metric between the predictions from our model $\widehat{y}$ and $c$, for each of the hierarchical levels. The leaf level predictions are trivially coherent. We find that our proposed model consistently produces more coherent predictions compared to both the models, indicating that our hierarchical regularization indeed encourages coherency in predictions, in addition to improving accuracy.

**Basis Visualization.** We visualize the basis generated by the BD model for the M5 validation set in Figure 3 (left). We notice that the bases capture various global temporal patterns in the dataset. In particular, most of the bases have a period of 7, indicating that they represent weekly patterns. We also show the predictions made from the various components of our model at all hierarchical levels, in Figure 3 (right). We notice that the predictions from the BD part closely resemble the general patterns of the true time series values, where as the AR model adds further adjustments to the predictions, including a constant bias, for most time series. For the leaf level (Level 3) predictions however, the final prediction is dominated by the AR model indicating that global temporal patterns may be less useful in this case.

## 7 CONCLUSION

In this paper, we proposed a method for hierarchical time series forecasting, consisting of two components, the TVAR model, and the BD model. The TVAR model is coherent by design, whereas we regularize the BD model to impose approximate coherency. Our model is fully differentiable and is trainable via SGD, while also being scalable with respect to the number of nodes in the hierarchy. Furthermore, it also does not require any additional pre-processing steps.

We empirically evaluated our method on three benchmark datasets and showed that our model consistently improved over state of the art baselines for most levels of the hierarchy. We perform an ablation study to justify the important components of our model and also show empirically that our forecasts are more coherent than the RNN baseline. Lastly, we also visualize the learned basis and observe that they capture various global temporal patterns.

In this work, we aimed at maximizing the overall performance without emphasizing performance at individual hierarchical levels. For future work, we plan to treat this is as a multi-objective problem with the aim of understanding the performance tradeoffs at various levels of the hierarchy. Finally, we also plan to extend our current model to probabilistic forecasting.

**Reproducibility statement:** We provide code for our model along with instructions in the supplementary file. The instructions are provided in the `readme.txt` file inside the zipped folder. Our code includes links to public datasets and code for preprocessing them. We also fix the random seed in our code in order for it to be reproducible. However, exact results may depend on the version of the python libraries, type of GPU and compute used.

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

## A    THEORY

### A.1    SUPPORT RECOVERY

**Lemma 1.** *Suppose $\boldsymbol{B}$ satisfies the lower eigenvalue condition (Assumption 1 in Appendix A.3) with parameter $C_{\min}$ and the mutual incoherence condition (Assumption 2 in Appendix A.3) with parameter $\gamma$. Also assume that the columns of the basis pool $\bar{\boldsymbol{B}}$ are normalized so that $\max_{j \in S^c} \|\bar{\boldsymbol{B}}^{(j)}\| \leq \sqrt{T}$, and the true parameter $\theta_0$ of the root satisfies*

$$\|\theta_0\|_\infty \geq \lambda_L \left[ \left\| \Sigma^{-1} \right\|_\infty + \frac{4\sigma}{\sqrt{L C_{\min}}} \right], \tag{9}$$

*where $\|A\|_\infty = \max_i \sum_j |A_{ij}|$ denotes matrix operator $\ell_\infty$ norm, and $\Sigma = \boldsymbol{B}^T \boldsymbol{B}/T$ denotes the empirical covariance matrix. Then for $\lambda_L \geq \frac{2}{\gamma} \sqrt{\frac{2\sigma^2 \log d}{LT}}$, with probability least $1 - 4\exp(-c_1 T \lambda^2)$ (for some constant $c_1$), the support $\widehat{S} = \{i \mid |\widehat{\alpha}_0| > 0\}$ recovered from the Lasso solution (see Algorithm 1) is equal to the true support $S$.*

*Proof.* We are given a pool of basis vectors $\bar{\boldsymbol{B}}$ from which the observed data is generated using a subset of $K$ columns which we have denoted by $\boldsymbol{B}$ in the text. We denote the correct subset of columns by $S$ and recover them from the observed data using basis pursuit - also known as the support recovery problem in the literature. Given the observed data and the pool of basis vectors $\bar{\boldsymbol{B}}$, we recover the support from the following regression problem corresponding to the root node time series.

$$y_0 = \bar{\boldsymbol{B}} \alpha + w_0, \quad w_0 \sim \mathcal{N}(\boldsymbol{0}, \sigma^2 \boldsymbol{I}/L), \tag{10}$$

where $\alpha$ is $K$-sparse with the non-zero indices at $S$, and the non-zero values equal $\theta_0$ - the true parameters of the root node. Here we have used the fact that the root node has a $1/L$ times smaller variance due to aggregation. The true support $S$ can be recovered from the observed data $y_0$, by solving the sparse regression problem (Lasso) given in Algorithm 2. A number of standard Lasso assumptions are needed to ensure that the support is identifiable, and that the non-zero parameters are large enough to be estimated. Assuming that $\bar{\boldsymbol{B}}_S (= \boldsymbol{B})$ and $\alpha$ satisfy all the assumptions of Theorem 2, the theorem ensures that the true support $S$ is recovered with high probability.  □

### A.2    PROOF OF THEOREM 1 - ERROR BOUNDS FOR REGULARIZED ESTIMATORS

For this proof, we assume that the support $S$ is recovered and the true basis functions $\boldsymbol{B}$ are known with high probability (see Section A.1). We divide the proof into multiple steps.

**Step I:**    By Corollary 1, the OLS estimate $\widehat{\theta}_0$ (see Algorithm 2) of parameters of the root node and the OLS estimate $\widetilde{\theta}_n$ (see Eq. (7)) can be bounded as,

$$\mathbb{E}[\|\widehat{\theta}_0 - \theta_0\|_\Sigma^2] \leq \frac{\sigma^2 K}{TL}, \quad \mathbb{E}[\|\widetilde{\theta}_n - \theta_n\|_\Sigma^2] \leq \frac{\sigma^2 K}{T} \quad \forall n \in \mathcal{L}(0). \tag{11}$$

**Step II:**    Next, using change of variables, we notice that the ridge regression loss for the child nodes (see Algorithm 2) is equivalent to the following.

$$\widehat{\psi}_n = \underset{\psi_n}{\operatorname{argmin}} \frac{1}{T} \|y_n - \boldsymbol{B}\widehat{\theta}_0 - \boldsymbol{B}\psi_n\|_2^2 + \lambda \|\psi_n\|_2^2 \quad \forall n \in \mathcal{L}(0), \tag{12}$$

where $\psi_n = \theta_n - \widehat{\theta}_0$. The final estimate for the child parameters can be written as a sum of the $\psi_n$ estimate and the root node estimate, $\widehat{\theta}_n = \widehat{\psi}_n + \widehat{\theta}_0$. We also consider a related problem that will help us in computing the errors bounds.

$$\widetilde{\psi}_n = \underset{\psi_n}{\operatorname{argmin}} \frac{1}{T} \|y_n - \boldsymbol{B}\theta_0 - \boldsymbol{B}\psi_n\|_2^2 + \lambda \|\psi_n\|_2^2 \quad \forall n \in \mathcal{L}(0). \tag{13}$$

Here we have replaced $\widehat{\theta}_0$ with the true value $\theta_0$. Note that this regression problem cannot be solved in practice since we do not have access to the true value of $\theta_0$. We will only use it to assist in the

analysis. Now we will bound the difference between the estimates $\widehat{\psi}_n$ and $\widetilde{\psi}_n$. The closed form solution for ridge regression is well known in the literature.

$$
\begin{aligned}
\widehat{\psi}_n &= T^{-1}(\Sigma + \lambda\boldsymbol{I})^{-1}\boldsymbol{B}^T(y_n - \boldsymbol{B}\widehat{\theta}_0) \\
\widetilde{\psi}_n &= T^{-1}(\Sigma + \lambda\boldsymbol{I})^{-1}\boldsymbol{B}^T(y_n - \boldsymbol{B}\theta_0),
\end{aligned}
$$

where $\Sigma = \boldsymbol{B}^T\boldsymbol{B}/T$, as defined earlier. The norm of the difference of the estimates can be bounded as

$$
\begin{aligned}
\widehat{\psi}_n - \widetilde{\psi}_n &= T^{-1}(\Sigma + \lambda\boldsymbol{I})^{-1}\boldsymbol{B}^T\boldsymbol{B}(\widetilde{\theta}_0 - \widehat{\theta}_0) \\
\implies \|\widehat{\psi}_n - \widetilde{\psi}_n\|_\Sigma^2 &= (\widetilde{\theta}_0 - \widehat{\theta}_0)^T\Sigma(\Sigma + \lambda\boldsymbol{I})^{-1}\Sigma(\Sigma + \lambda\boldsymbol{I})^{-1}\Sigma(\widetilde{\theta}_0 - \widehat{\theta}_0) \\
&= (\widetilde{\theta}_0 - \widehat{\theta}_0)^T\Sigma(\Sigma + \lambda\boldsymbol{I})^{-1}\Sigma(\Sigma + \lambda\boldsymbol{I})^{-1}\Sigma(\widetilde{\theta}_0 - \widehat{\theta}_0) \\
&= (\widetilde{\theta}_0 - \widehat{\theta}_0)^TVD\left[\frac{\lambda_i^3}{(\lambda_i + \lambda)^2}\right]V^T(\widetilde{\theta}_0 - \widehat{\theta}_0).
\end{aligned}
$$

Here we have used an eigen-decomposition of the symmetric sample covariance matrix as $\Sigma = VD[\lambda_i]V^T$. We use the notation $D[\lambda_i]$ to denote a diagonal matrix with values $\lambda_i$ on the diagonal. The above can be further upper bounded using the fact that $\lambda_i \leq \lambda_i + \lambda$.

$$
\|\widehat{\psi}_n - \widetilde{\psi}_n\|_\Sigma^2 \leq (\widetilde{\theta}_0 - \widehat{\theta}_0)^TVD[\lambda_i]V^T(\widetilde{\theta}_0 - \widehat{\theta}_0) = \|\widetilde{\theta}_0 - \widehat{\theta}_0\|_\Sigma^2. \tag{14}
$$

**Step III:** Now we will bound the error on $\widetilde{\psi}_n$ and finally use it in the next step with triangle inequality to prove our result. Note that $y_n - \boldsymbol{B}\theta_0 = \boldsymbol{B}(\theta_n - \theta_0) + w_n$. Therefore, we can see from Eq. (13) that $\widetilde{\psi}_n$ is an estimate for $\theta_n - \theta_0$. Using the fact that $\|\theta_n - \theta_0\|_2 \leq \beta$ and corollary 1, $\widetilde{\psi}_n$ can be bounded as,

$$
\mathbb{E}[\|\widetilde{\psi} - (\theta_n - \theta_0)\|_\Sigma^2] \leq \frac{r^2\beta^2\sigma^2 K}{Tr^2\beta^2 + \sigma^2 K}. \tag{15}
$$

Finally using triangle inequality, we bound the error of our estimate $\widehat{\theta}_n$.

$$
\begin{aligned}
\|\widehat{\theta}_n - \theta_n\|_\Sigma^2 &= \|\widehat{\psi}_n + \widehat{\theta}_0 - \theta_n\|_\Sigma^2 \quad \text{(Using the decomposition from Step II)}. \\
&\leq 3\left(\|\widehat{\psi}_n - \widetilde{\psi}_n\|_\Sigma^2 + \|\widetilde{\psi}_n - (\theta_n - \theta_0)\|_\Sigma^2 + \|\widehat{\theta}_0 - \theta_0\|_\Sigma^2\right) \\
&\quad \text{(Using triangle and Cauchy-Schwartz inequality)} \\
&\leq 3\left(\|\widetilde{\psi}_n - (\theta_n - \theta_0)\|_\Sigma^2 + 2\|\widehat{\theta}_0 - \theta_0\|_\Sigma^2\right) \quad \text{(Using Eq. (14))}.
\end{aligned}
$$

Taking the expectation of the both sides, and using Eq. (11) and (15), we get the desired result.

$$
\mathbb{E}\|\widehat{\theta}_n - \theta_n\|_\Sigma^2 \leq 3\frac{r^2\beta^2\sigma^2 K}{Tr^2\beta^2 + \sigma^2 K} + 6\frac{\sigma^2 K}{TL}.
$$

## A.3 Review of Sparse Linear Regression

We consider the following sparse recovery problem. We are given data $(\boldsymbol{X}, y) \in \mathbb{R}^{n\times d} \times \mathbb{R}^n$ following the observation model $y = \boldsymbol{X}\theta^* + w$, where $w \sim \mathcal{N}(\boldsymbol{0}, \sigma^2\boldsymbol{I})$, and $\theta^*$ is supported in the indices indexed by a set $S$ ($S$-sparse). We estimate $\theta^*$ using the following Lagrangian Lasso program,

$$
\widehat{\theta} \in \underset{\theta \in \mathbb{R}^n}{\operatorname{argmin}}\left\{\frac{1}{2n}\|y - \boldsymbol{X}\theta\|_2^2 + \lambda_n\|\theta\|_1\right\} \tag{16}
$$

We consider the fixed design setting, where the matrix $\boldsymbol{X}$ is fixed and not sampled randomly. Following (Wainwright, 2009), we make the following assumptions required for recovery of the true support $S$ of $\theta^*$.

**Assumption 1** (Lower eigenvalue). *The smallest eigenvalue of the sample covariance sub-matrix indexed by $S$ is bounded below:*

$$
\Lambda_{\min}\left(\frac{\boldsymbol{X}_S^T\boldsymbol{X}_S}{n}\right) \geq C_{\min} > 0 \tag{17}
$$

**Assumption 2** (Mutual incoherence). *There exists some $\gamma \in (0, 1]$ such that*

$$\left\|\!\left\| \boldsymbol{X}_{S^c}^T \boldsymbol{X}_S (\boldsymbol{X}_S^T \boldsymbol{X}_S)^{-1} \right\|\!\right\|_\infty \leq 1 - \gamma, \tag{18}$$

*where $\|\!\|A\|\!\|_\infty = \max_i \sum_j |A_{ij}|$ denotes matrix operator $\ell_\infty$ norm.*

**Theorem 2** (Support Recovery, Wainwright (2009)). *Suppose the design matrix satisfies assumptions 1 and 2. Also assume that the design matrix has its $n$-dimensional columns normalized so that $\max_{j \in S^c} \|X_j\|_2 \leq \sqrt{n}$. Then for $\lambda_n$ satisfying,*

$$\lambda_n \geq \frac{2}{\gamma} \sqrt{\frac{2\sigma^2 \log d}{n}}, \tag{19}$$

*the Lasso solution $\widehat{\theta}$ satisfies the following properties with a probability of at least $1 - 4\exp(-c_1 n\lambda_n^2)$:*

1. *The Lasso has a unique optimal solution $\widehat{\theta}$ with its support contained within the true support $S(\widehat{\theta}) \subseteq S(\theta^*)$ and satisfies the $\ell_\infty$ bound*

$$\|\widehat{\theta}_S - \theta_S^*\|_\infty \leq \underbrace{\lambda_n \left[ \left\|\!\left\| \left( \frac{\boldsymbol{X}_S^T \boldsymbol{X}_S}{n} \right)^{-1} \right\|\!\right\|_\infty + \frac{4\sigma}{\sqrt{C_{\min}}} \right]}_{g(\lambda_n)}, \tag{20}$$

2. *If in addition, the minimum value of the regression vector $\theta^*$ is lower bounded by $g(\lambda_n)$, then it recovers the exact support.*

## A.4 REVIEW OF RIDGE REGRESSION

In this section we review the relevant background from (Hsu et al., 2012) on fixed design ridge regression. As usual, we assume data $(\boldsymbol{X}, y) \in \mathbb{R}^{n \times d} \times \mathbb{R}^n$ following the observation model $y = \boldsymbol{X}\theta^* + w$, where $w \sim \mathcal{N}(\boldsymbol{0}, \sigma^2 \boldsymbol{I})$. Define the *ridge estimator* $\widehat{\theta}$ as the minimizer of the $\ell_2$ regularized mean squared error,

$$\widehat{\theta} \in \operatorname*{argmin}_{\theta \in \mathbb{R}^n} \left\{ \frac{1}{n} \|y - \boldsymbol{X}\theta\|_2^2 + \lambda\|\theta\|_2^2 \right\} \tag{21}$$

We denote the sample covariance matrix by $\Sigma = \boldsymbol{X}^T \boldsymbol{X}/n$. Then for any parameter $\theta$, the expected $\ell_2$ prediction error is given by, $\|\theta - \theta^*\|_\Sigma^2 = \|\boldsymbol{X}(\theta - \theta^*)\|_2^2/n$. We also assume the standard ridge regression setting of bounded $\|\theta^*\|_2 \leq B$. We have the following proposition from Hsu et al. (2012) on expected error bounds for ridge regression.

**Proposition 1** (Hsu et al. (2012)). *For any regularization parameter $\lambda > 0$, the expected prediction loss can be upper bounded as*

$$\mathbb{E}[\|\widehat{\theta} - \theta^*\|_\Sigma^2] \leq \sum_j \frac{\lambda_j}{(\lambda_j/\lambda + 1)^2} \theta_j^{*2} + \frac{\sigma^2}{n} \sum_j \left( \frac{\lambda_j}{\lambda_j + \lambda} \right)^2, \tag{22}$$

*where $\lambda_i$ denote the eigenvalues of the empirical covariance matrix $\Sigma$.*

Using the fact that $\lambda_j \leq \mathbf{tr}\,(\Sigma)$, and $x/(x+c)$ is increasing in $x$ for $x \geq 0$, the above bound can be simplified as,

$$
\begin{aligned}
\mathbb{E}[\|\widehat{\theta} - \theta^*\|_\Sigma^2] &\leq \frac{\mathbf{tr}\,(\Sigma)}{(\mathbf{tr}\,(\Sigma)/\lambda + 1)^2} \|\theta^*\|^2 + \frac{\sigma^2 d}{n} \left( \frac{\mathbf{tr}\,(\Sigma)}{\mathbf{tr}\,(\Sigma) + \lambda} \right)^2 \\
&\leq \frac{\mathbf{tr}\,(\Sigma) B^2 \lambda^2 + \mathbf{tr}\,(\Sigma)^2 \sigma^2 d/n}{(\mathbf{tr}\,(\Sigma) + \lambda)^2}
\end{aligned}
$$

Assuming that the covariate vectors $\boldsymbol{X}_i$ are norm bounded as $\|\boldsymbol{X}_i\|_2 \leq r$, and using the fact that $\mathbf{tr}\,(\Sigma) \leq r^2$, gives us the following corollary.

**Corollary 1.** *When choosing $\lambda = \frac{\sigma^2 d}{nB^2}$, the prediction loss can be upper bounded as,*

$$\mathbb{E}[\|\widehat{\theta} - \theta^*\|_\Sigma^2] \leq \frac{r^2 B^2 \sigma^2 d}{nr^2 B^2 + \sigma^2 d}. \tag{23}$$

*The usual ordinary least squares bound of $\frac{\sigma^2 d}{n}$ can be derived when considering the limit $B \to \infty$, corresponding to $\lambda = 0$.*

## A.5 FURTHER DISCUSSION ON THEOREM 1

**Upper bound $\|\theta_n - \theta_0\|_2 \leq \beta$:** The upper bound $\beta$ essentially bounds the distance between sibling leaf embeddings belonging to the same parent. This is directly related to an upper bound on the distance between the parent embedding $\theta_0$ and the leaf embeddings $\theta_n$, as $\theta_n$ is essentially the mean of the leaf nodes (mean property). In many practical scenarios, the children time series of a parent may not have too different seasonal trends (for example power consumption of houses in the same neighborhood, or sales of items under the same category) resulting in the parent time series following similar trends as well.

**Bounding $\|\theta_n - \theta_0\|_2$:** In most theoretical analyses of linear regression (Wainwright, 2019), the main quantity of interest is the prediction error $\|\theta_n - \theta_0\|_\Sigma$ rather than the parameter estimation error $\|\theta_n - \theta_0\|_2$, as the former is directly related to the performance metric of the model. However, a bound on the parameter estimation error can be easily established using the property that $\|\theta_n - \theta_0\|_2 \leq \|\theta_n - \theta_0\|_\Sigma / \sqrt{C_{\min}}$, where $C_{\min}$ is the lower bound on the smallest eigenvalue of the sample covariance matrix as defined in Assumption 1 in Section A.3.

## B FURTHER EXPERIMENTAL DETAILS

### B.1 ACCURACY METRICS

In this section we define the evaluation metrics used in this paper. Denote the true values by $\boldsymbol{y}$ and the predicted values by $\widehat{\boldsymbol{y}}$, both $n$-dimensional vectors.

1. Symmetric mean absolute percent error SMAPE $= \frac{2}{n} \sum_i \frac{|\widehat{\boldsymbol{y}}_i - \boldsymbol{y}_i|}{|\boldsymbol{y}_i| + |\widehat{\boldsymbol{y}}_i|}$.

2. Weighted absolute percentage error WAPE $= \frac{\sum_i |\widehat{\boldsymbol{y}}_i - \boldsymbol{y}_i|}{\sum_i |\boldsymbol{y}_i|}$.

### B.2 DATASET DETAILS

We use three publicly available benchmark datasets for our experiments.

1. The M5 dataset[3] consists of time series data of product sales from 10 Walmart stores in three US states. The data consists of two different hierarchies: the product hierarchy and store location hierarchy. For simplicity, in our experiments we use only the product hierarchy consisting of 3k nodes and 1.8k time steps. The validation scores are computed using the predictions from time steps 1843 to 1877, and test scores on steps 1878 to 1913.

2. The Favorita dataset[4] is a similar dataset, consisting of time series data from Corporación Favorita, a South-American grocery store chain. As above, we use the product hierarchy, consisting of 4.5k nodes and 1.7k time steps. The validation scores are computed using the predictions from time steps 1618 to 1652, and test scores on steps 1653 to 1687.

3. The Australian Tourism dataset[5] consists of monthly domestic tourist count data in Australia across 7 states which are sub-divided into regions, sub-regions, and visit-type. The data consists of around 500 nodes and 230 time steps. The validation scores are computed using the predictions from time steps 122 to 156, and test scores on steps 157 to 192.

For the three datasets, all the time-series (corresponding to both leaf and higher-level nodes) of the hierarchy that we used are present in the training data.

### B.3 TRAINING DETAILS

Table 3 presents all the hyperparameters used in our proposed model. All models were trained via SGD using the Adam optimizer (Kingma & Ba, 2014), and the training data was standardized to mean zero and unit variance. The datasets were split into train, validation, and test sets, the sizes of which

---

[3] https://www.kaggle.com/c/m5-forecasting-accuracy/
[4] https://www.kaggle.com/c/favorita-grocery-sales-forecasting/
[5] https://robjhyndman.com/publications/mint/

Table 3: Final model hyperparameters for various datasets tuned using the Mean WAPE metric on the validation set.

| Model hyperparameters | M5 | Favorita | Tourism |
|---|---|---|---|
| LSTM hidden dim | 42 | 24 | 14 |
| Embedding dim $K$ | 8 | 8 | 6 |
| NMF rank $R$ | 12 | 4 | 6 |
| Multi-Horizon decoder hidden dim | 24 | 16 | 12 |
| Embedding regularization $\lambda_E$ | 3.4e-6 | 4.644e-4 | 7.2498e-8 |
| History length $H$ and forecast horizon $F$ | (28, 7) | (28, 7) | (24, 4) |
| No. of rolling val/test windows | 5 | 5 | 3 |
| Initial learning rate | 0.004 | 0.002 | 0.07 |
| Decay rate and decay interval | (0.5, 6) | (0.5, 6) | (0.5, 6) |
| Early stopping patience | 10 | 10 | 10 |
| Training epochs | 40 | 40 | 40 |
| Batch size | 512 | 512 | 512 |
| Total #params | 80k | 80k | 8k |

are given in the Table 3. We used learning rate decay and early stopping using the Mean WAPE score on the validation set, with a patience of 10 for all models. We tuned the model hyper-parameters using the same metric. The various model hyper-parameters are given in Table 3.

All our experiments were implemented in Tensorflow 2, and run on a Titan Xp GPU with 12GB of memory. The computing server we used, had 256GB of memory, and 32 CPU cores, however, our code did not seem to use more than 10GB of memory and 4 CPU cores.

**Mini-Batching:** During each training iteration, we sample a minibatch of time series for computing the gradients. For constructing a minibatch, first a time window (of length $H + F$) is uniformly randomly selected from the training time steps, after which a subset of nodes is sampled from the hierarchy tree. The time series data corresponding to this subset of nodes and the sampled time window constitutes our minibatch.

**Predicting for new time steps:** Once trained, our model can be used to predict for new datapoints without any retraining. However, in practice it may be beneficial to retrain the model with newer data to improve performance, even though this is not a constraint for our proposed approach.

**Knowledge of the hierarchy:** Our proposed approach requires the hierarchy to be known during training to be able to regularize the embeddings. In scenarios where new aggregations are presented during test time, a reasonable prediction can be produced using the following strategy: the embedding of the new aggregation is set to the mean of the embeddings of nodes in that aggregation - the rest of the model remains the same. Predictions for unseen aggregations are made in the same way as with seen aggregations. This idea is beyond the scope of the current paper, thus left for future exploration.

### B.4 FURTHER DETAILS ABOUT GLOBAL STATE $Z$

In many practical scenarios, the evolving global state of the set of time series may not be captured by the global covariates $X$ only. For instance, when there is an overall increase/decrease in sales across all time series, it is captured in the past values $Y$ rather than $X$. As a result, it may be required to feed in past values of time series to the BD model. However, since we cannot feed in the whole set of time series (order of 1000s) without leading to scalability issues, we choose a small set of representative time series using Non-negative Matrix Factorization, thus approximating the global state. NMF assumes that the columns of the time series matrix $Y \in \mathbb{R}^{T \times N}$ lies in the convex set spanned by a small *subset* of columns. However, this assumption may not hold true for many datasets, and therefore, most NMF algorithms compute an approximate factorization. We use this aforementioned subset of columns as an approximation to the global state. We would also like to

emphasize that our prediction model only sees the past values $Z_\mathcal{H}$ as input since the future time series values are unknown.

While NMF is one of the choices for $Z$, it is definitely not the only choice. Another option, PCA (or equivalently SVD), may lead to latent vectors which do not have any temporal dependencies thus requiring additional temporal regularization (Sen et al., 2019; Wilson et al., 2008). One may also use more sophisticated methods such as Temporal Latent Auto-Encoders (Nguyen & Quanz, 2021). We leave this idea for future exploration.

### B.5 BASELINES

Further details about the baselines we compare with are provided as below. For a fair comparison, we use the the same Mean Absolute Error loss functions for training, and ensure that all the baseline models have similar number of parameters.

1. *RNN*: We use an LSTM decoder and encoder to implement a seq-2-seq model shared across all the time series, trained using mean absolute error loss.

2. *DeepGLO* (Sen et al., 2019): We use the implementation released by the authors on Github[6]. We modify the loss function, data handling and evaluation to adapt to our setting.

3. *DCRNN* (Li et al., 2017): DCRNN being a GNN based approach requires a *correlation* graph as input. We use the official implementation released by the authors[7] and provide the hierarchy tree as the input graph. The implementation uses MAE loss by default.

4. *Deep Factors (DF)* (Wang et al., 2019): The original implementation released by the authors makes rolling probabilistic forecasts. We implement our own version in Tensorflow using an LSTM encoder for the global model (producing point predictions) as in the original implementation, while leaving out the probabilistic local model. We manually tune the hyper-parameters for each of the datasets on the validation set.

5. *$L_2$Emb* (Gleason, 2020): We implement this model using an LSTM decoder and encoder with MAE as the main training loss. In addition, we also use node embeddings which are fed as input to the encoder and decoder, and regularized according to the hierarchy as described by Gleason (2020).

6. *SHARQ* (Han et al., 2021): We were not able to find an official release of SHARQ. We implemented it using an LSTM based seq-to-seq model, with layer-wise training as described in the paper. We used MAE as the *data fit loss* function and the default squared error *reconciliation loss* regularizer (See Han et al. (2021) for terminology).

7. *HierE2E* (Rangapuram et al., 2021): The official implementation produces rolling probabilistic forecasts. We adapt it to point forecasts by using the proposed projection step on outputs from a seq-2-seq model.

8. *RNN+ERM* (Ben Taieb & Koo, 2019): This approach involves learning a sparse projection matrix from data, resulting in coherent predictions. We tune the sparsity parameter using the model accuracy on the validation set.

### B.6 MORE RESULTS

Table 4 show the test metrics averaged over 10 independent runs on the three datasets along with the standard deviations.

---

[6] https://github.com/rajatsen91/deepglo
[7] https://github.com/liyaguang/DCRNN/

Table 4: WAPE/SMAPE test metrics for all the three datasets, averaged over 10 runs. The standard deviations are shown in the parenthesis. We bold the smallest mean in each column and anything that comes within two standard deviations.

| M5 | Level 0 | Level 1 | Level 2 | Level 3 | Mean |
|---|---|---|---|---|---|
| HiReD | **0.048** / **0.048** (0.0011) / (0.0011) | **0.055** / **0.053** (0.0006) / (0.0006) | **0.072** / **0.077** (0.0007) / (0.0006) | 0.279 / 0.511 (0.0003) / (0.0012) | **0.113** / **0.172** (0.0005) / (0.0006) |
| RNN | 0.059 / 0.059 (0.002) / (0.003) | 0.083 / 0.083 (0.013) / (0.011) | 0.085 / 0.098 (0.002) / (0.004) | 0.282 / 0.517 (0.006) / (0.007) | 0.127 / 0.189 (0.005) / (0.005) |
| DF | 0.055 / 0.056 (0.001) / (0.001) | 0.061 / 0.060 (0.001) / (0.001) | 0.076 / 0.085 (0.001) / (0.002) | 0.272 / **0.501** (0.000) / (0.002) | 0.116 / 0.176 (0.001) / (0.001) |
| DeepGLO | 0.077 / 0.081 (0.0003) / (0.0004) | 0.087 / 0.092 (0.0003) / (0.0004) | 0.099 / 0.113 (0.0003) / (0.0003) | 0.278 / 0.538 (0.0001) / (0.0001) | 0.135 / 0.206 (0.0003) / (0.0003) |
| DCRNN | 0.078 / 0.079 (0.006) / (0.007) | 0.096 / 0.092 (0.005) / (0.004) | 0.165 / 0.193 (0.003) / (0.007) | 0.282 / 0.512 (0.000) / (0.000) | 0.156 / 0.219 (0.002) / (0.003) |
| $L_2$Emb | 0.055 / 0.056 (0.0016) / (0.001) | 0.064 / 0.063 (0.0014) / (0.001) | 0.080 / 0.092 (0.0011) / (0.001) | **0.269** / **0.501** (0.0003) / (0.003) | 0.117 / 0.178 (0.0009) / (0.001) |
| SHARQ | 0.093 / 0.096 (0.002) / (0.002) | 0.071 / 0.062 (0.004) / (0.003) | 0.099 / 0.094 (0.002) / (0.001) | 0.277 / 0.528 (0.000) / (0.000) | 0.135 / 0.195 (0.001) / (0.001) |
| RNN+ERM | 0.052 / 0.052 (0.001) / (0.001) | 0.066 / 0.071 (0.001) / (0.002) | 0.084 / 0.104 (0.001) / (0.002) | 0.286 / 0.520 (0.002) / (0.004) | 0.122 / 0.187 (0.001) / (0.001) |
| Hier-E2E | 0.152 / 0.160 (0.002) / (0.002) | 0.152 / 0.158 (0.002) / (0.002) | 0.152 / 0.181 (0.002) / (0.002) | 0.396 / 0.615 (0.001) / (0.002) | 0.213 / 0.278 (0.002) / (0.002) |
| Hier-E2E Large | **0.047** / 0.050 (0.002) / (0.003) | 0.057 / 0.063 (0.001) / (0.001) | **0.067** / 0.080 (0.001) / (0.001) | 0.347 / 0.573 (0.001) / (0.001) | 0.130 / 0.192 (0.001) / (0.001) |

| Favorita | Level 0 | Level 1 | Level 2 | Level 3 | Mean |
|---|---|---|---|---|---|
| HiReD | 0.061 / **0.061** (0.002) / (0.002) | **0.094** / **0.182** (0.001) / (0.002) | **0.127** / **0.267** (0.001) / (0.003) | 0.210 / **0.322** (0.000) / (0.004) | **0.123** / **0.208** (0.001) / (0.002) |
| RNN | 0.067 / 0.068 (0.004) / (0.003) | 0.114 / 0.197 (0.003) / (0.004) | 0.134 / 0.290 (0.002) / (0.005) | **0.203** / 0.339 (0.001) / (0.005) | 0.130 / 0.223 (0.002) / (0.004) |
| DF | 0.064 / 0.064 (0.003) / (0.004) | 0.110 / 0.194 (0.002) / (0.003) | 0.135 / 0.291 (0.002) / (0.007) | 0.213 / 0.343 (0.001) / (0.007) | 0.130 / 0.223 (0.002) / (0.004) |
| DeepGLO | 0.098 / 0.088 (0.001) / (0.001) | 0.126 / 0.197 (0.001) / (0.001) | 0.156 / 0.338 (0.001) / (0.001) | 0.226 / 0.404 (0.001) / (0.001) | 0.151 / 0.256 (0.001) / (0.001) |
| DCRNN | 0.080 / 0.080 (0.004) / (0.005) | 0.120 / 0.212 (0.001) / (0.002) | 0.134 / 0.328 (0.000) / (0.000) | **0.204** / 0.389 (0.000) / (0.000) | 0.134 / 0.252 (0.001) / (0.001) |
| $L_2$Emb | 0.070 / 0.070 (0.003) / (0.003) | 0.114 / 0.199 (0.002) / (0.004) | 0.136 / 0.276 (0.001) / (0.006) | 0.207 / **0.321** (0.001) / (0.007) | 0.132 / 0.216 (0.002) / (0.004) |
| SHARQ | 0.088 / 0.085 (0.002) / (0.002) | 0.142 / 0.199 (0.001) / (0.001) | 0.156 / 0.335 (0.001) / (0.001) | 0.230 / 0.404 (0.000) / (0.000) | 0.154 / 0.256 (0.000) / (0.000) |
| RNN+ERM | **0.056** / **0.058** (0.002) / (0.002) | 0.103 / 0.185 (0.001) / (0.003) | 0.129 / 0.283 (0.001) / (0.005) | 0.220 / 0.348 (0.001) / (0.005) | 0.127 / 0.219 (0.001) / (0.003) |
| Hier-E2E | 0.120 / 0.125 (0.005) / (0.006) | 0.206 / 0.334 (0.003) / (0.005) | 0.247 / 0.448 (0.002) / (0.006) | 0.409 / 0.573 (0.007) / (0.014) | 0.245 / 0.370 (0.004) / (0.007) |
| Hier-E2E Large | 0.082 / 0.077 (0.002) / (0.002) | 0.168 / 0.263 (0.003) / (0.010) | 0.190 / 0.360 (0.002) / (0.003) | 0.314 / 0.440 (0.002) / (0.001) | 0.189 / 0.285 (0.002) / (0.003) |

| Tourism | Level 0 | Level 1 | Level 2 | Level 3 | Level 4 | Mean |
|---|---|---|---|---|---|---|
| HiReD | **0.059** / **0.061** (0.001) / (0.001) | **0.125** / 0.162 (0.001) / (0.003) | **0.172** / 0.225 (0.001) / (0.002) | **0.229** / 0.376 (0.001) / (0.004) | **0.347** / **0.786** (0.001) / (0.007) | **0.186** / **0.322** (0.001) / (0.002) |
| RNN | 0.110 / 0.106 (0.001) / (0.001) | 0.148 / 0.164 (0.001) / (0.002) | 0.188 / 0.231 (0.001) / (0.001) | 0.240 / 0.385 (0.000) / (0.006) | 0.369 / **0.782** (0.001) / (0.012) | 0.211 / 0.333 (0.001) / (0.002) |
| DF | 0.097 / 0.096 (0.003) / (0.002) | 0.141 / 0.170 (0.002) / (0.002) | 0.187 / 0.240 (0.001) / (0.002) | 0.241 / 0.380 (0.000) / (0.014) | 0.355 / **0.783** (0.000) / (0.014) | 0.204 / 0.334 (0.001) / (0.003) |
| DeepGLO | 0.089 / 0.079 (0.0002) / (0.0002) | **0.126** / **0.158** (0.0001) / (0.0001) | 0.179 / **0.218** (0.0001) / (0.0001) | 0.234 / **0.372** (0.0001) / (0.0001) | 0.364 / 0.900 (0.0001) / (0.0002) | 0.199 / 0.346 (0.0001) / (0.0001) |
| DCRNN | 0.187 / 0.171 (0.003) / (0.003) | 0.231 / 0.248 (0.002) / (0.003) | 0.258 / 0.279 (0.001) / (0.002) | 0.293 / 0.398 (0.001) / (0.001) | 0.434 / 0.865 (0.000) / (0.000) | 0.281 / 0.392 (0.000) / (0.001) |
| $L_2$Emb | 0.114 / 0.115 (0.007) / (0.007) | 0.153 / 0.180 (0.002) / (0.004) | 0.192 / 0.244 (0.002) / (0.002) | 0.245 / 0.385 (0.001) / (0.002) | 0.372 / **0.789** (0.002) / (0.010) | 0.215 / 0.342 (0.002) / (0.003) |
| SHARQ | 0.100 / 0.104 (0.005) / (0.002) | 0.164 / 0.209 (0.002) / (0.001) | 0.217 / 0.260 (0.003) / (0.002) | 0.265 / 0.386 (0.003) / (0.001) | 0.399 / 0.931 (0.003) / (0.004) | 0.229 / 0.378 (0.001) / (0.001) |
| RNN+ERM | 0.078 / 0.079 (0.005) / (0.005) | 0.155 / 0.206 (0.003) / (0.006) | 0.225 / 0.291 (0.004) / (0.006) | 0.307 / 0.498 (0.006) / (0.008) | 0.488 / 1.013 (0.009) / (0.010) | 0.251 / 0.417 (0.005) / (0.006) |
| Hier-E2E | 0.110 / 0.113 (0.002) / (0.002) | 0.143 / 0.161 (0.002) / (0.003) | 0.187 / 0.232 (0.002) / (0.003) | 0.240 / 0.371 (0.001) / (0.004) | 0.358 / 0.824 (0.001) / (0.003) | 0.208 / 0.340 (0.001) / (0.002) |

