# OpenReview forum: "Hierarchically Regularized Deep Forecasting"
_ICLR.cc/2022/Conference — ICLR 2022 Submitted_

### Official Review · Reviewer_UnFM · 2021-11-01

**Correctness:** 3
**Technical Novelty And Significance:** 3
**Empirical Novelty And Significance:** 3
**Recommendation:** 6
**Confidence:** 3

**Main Review:**

The paper’s subject, forecasting, is being studied regularly in academia and
research groups, and hierarchical forecasting specifically, has many use
cases in industry.
The paper is well written and organized overall and conveys a good “reader
journey” experience. The figures and tables describing and emphasizing
relevant parts and have good explanations.
The ‘Introduction’ describing the problem, gives relevant use cases and
serves as a great entrance gate for the paper. Figure1 gives an important
visual intuition. The ‘Problem Statement’ paragraph is a must have in every
paper and helps to prepare the reader for the ‘Problem Setting’ section and
the suggested model section. For a weakness, I would like to point out that
the paper is dealing with ‘additive coherence constraints’ only, while one of
the use cases described in the first paragraph is financial predictions, that
might not have the additive constraint (for example in financial markets), but
a different form of aggregation. It’s fine that the paper doesn’t deal with it,
but it would be worth if the paper touched on this point.
In section 4, the ‘HiReD’ model is being very well explained and Figure2
completes the picture of the model perfectly. For the model itself, the
strength of it is the fact that it uses simple building blocks that can be
implemented easily with state-of-the-art gradient based libraries, also it can
deal with hierarchical structured data and scales linearly with more time
series. Moreover, it uses information from the entire hierarchy in training but
not in the forecasting phase, that should result in faster forecasts. Its
weakness is the fact that you must know the hierarchy structure in advance
(it’s OK for the use cases described in the paper, but it might not always be
the case). The predictions are point forecasts and not probabilistic, while in
many domains confidence level or intervals are very important for decision
making (like in retail). Another issue that wasn’t discussed is the scalability of
the model to new time steps – do you have to train the model all over again
with all history plus the new time step to produce an improved prediction (for
t+2)?

In the ‘Experiments’ section, the method is compared to a variety of
methods, that can be divided into: (i) neural network (NN) models for time
series - that are not necessarily optimized for hierarchical problems; (ii) graph
NN models; (iii) NN for hierarchical models; (iv) reconciliation models. The
explanations, metrics and result tables are well written, but I want to point
out that two methods that were introduced in the ‘Related Work on Deep
Hierarchical Models’ part, Han et al. (2021) and Yanchenko et al. (2021), were
not  used as baseline. As actionable feedback, I recommend including in
the baseline more NN for hierarchical models methods than “regular” NN
time series methods, as the main subject of the paper is hierarchical time
series forecasting.

**Summary Of The Paper:**

The paper introduces a method for hierarchical time series forecasting.
The problem setting is: given historical hierarchical univariate time series
data and given historical and future features (like holidays, etc..), try to
predict future values for all time series, while keeping the coherence
constraints of the hierarchy.
The forecasting is broken down into an autoregressive part with shared
parameters for all time series, and a basis decomposition with different
weights for each time series (regularized by the hierarchy structure).
The method is then being compared to different baseline methods on three
datasets.

**Summary Of The Review:**

The paper is well written and gives great introduction for the subject. The
suggested method’s strength is in its easy to implement building blocks, the
ability to take as into account information from all hierarchy and to scale well
for larger hierarchies, though it could be better if the issue of dealing with
new time steps would be addressed. The datasets chosen for the
experiments are showing good results against relevant baseline, although
there are few more relevant methods that were not included in the baseline.

---

> ### Author Response · Authors · 2021-11-15
> **Thanks for the feedback**
>
> We thank the reviewer for the detailed review and positive comments, especially on the clarity. We address the other concerns below:
> - Additive coherence constraints: Thank you for pointing this out. Indeed, our proposed method is strongly tied to additive coherence constraints only. Extending our ideas to other aggregation constraints is an interesting topic for future work. We will address this point in our draft once we are done exploring more papers on this topic. We are also happy to include any citations or pointers that you can provide.
> - Knowledge of the hierarchy: Yes our proposed approach does require the hierarchy during training to be able to regularize the embeddings. In cases where a hierarchy is not available, as long as we train using randomly constructed aggregations, we hope that our model will still be able to generalize to new aggregations during test time. In such scenarios, the embedding of a new aggregation should be set to the mean of the embeddings of nodes in that aggregation - the rest of the model will remain the same. While this is beyond the realm of the current work, we have added a discussion about this interesting direction in Appendix B.3.
> - Probabilistic prediction: We do agree that probabilistic prediction is important in many applications. However, similar to many prior works, in this paper we solely focus on point forecasts as a first step and leave the extension to probabilistic forecasting to future work.
> - Scalability to new time steps: Our model does not need to be re-trained for every new time step added. In fact, in our evaluation, we train the model only on the train set and use the exact same model on the test set for a rolling evaluation without any retraining. We have added a paragraph making this clear in Appendix B.3.
> - Experiments: Thanks for feedback about improving the experiments section. We will update these in our upcoming draft. We are working on running SHARQ by Han et. al. 2021, and will update the draft as soon as we have results. As for Yanchenko et. al. 2021, we weren’t able to find an implementation yet, but we will attempt to implement it on our own.
>
> Please let us know if there are any more concerns.

---

> ### Author Response · Authors · 2021-11-22
> **Update on experiments**
>
> We have now added experimental results (Table 1) for a new baseline SHARQ (Han et. al. 2021). We were not able to find any publicly available implementations for both SHARQ and Yanchenko et. al. 2021. We implemented our own version of SHARQ as it was relatively straightforward to do so. We could not find any reference implementations for Yanchenko et. al. 2021 to build up on. Implementing it from scratch does not seem to be straightforward.
>
> Please let us know if any major concerns remain.

---

### Official Review · Reviewer_iRbQ · 2021-11-02

**Correctness:** 3
**Technical Novelty And Significance:** 3
**Empirical Novelty And Significance:** 2
**Recommendation:** 5
**Confidence:** 5

**Main Review:**

Strengths:
- Dimension reduction is an important topic that has been overlooked in the literature on hierarchical forecasting.
- Experiments with several publicly-available real-world datasets.

Weaknesses:
- Paper clarity should be improved. Reading the paper raised a lot of (unanswered) questions. See comments below.
- About the proposed method:
* The proposed method produces roughly/approximate coherent forecasts. It is hard to compare methods that produce coherent and incoherent forecasts.
* The paper only considers point forecasting. Probabilistic forecasting is a more challenging and important problem.
* It is not clear whether the basis decomposition should be applied on all series. A hierarchical time series is a multivariate time series with hierarchical aggregation constraints. As a result, some series are linear combinations of other series. Isn't it enough to just apply it to the bottom-level series?
- I am wondering if the comparison between methods is fair. In fact, the proposed method minimizes (regularized) MAE while other baselines minimize MSE. Your method is more aligned with the evaluation metrics, WAPE/SMAPE, which are essentially scaled absolute errors.


Other comments:
- Related work: literature on hierarchical probabilistic forecasting should be discussed, as well as dimension reduction methods for multivariate time series.
- "Hierarchical coherency" has nothing to do with coherence as discussed in Thomson et al., 2019.
- Section 2 should be split in two different paragraphs. One on mean forecasting and the other on probabilistic/quantile forecasting. You cannot really compare these two types of methods.
- Figure 4: it is not clear why Tourism has a non-zero L3 colum.
- What does "asymptotically better" means? Please be more rigorous or avoid this term.
- "under the reasonable assumption of the parent embedding being close to all the children embeddings" -> Why is it a reasonable assumption?
- Why imposing the hierarchical constraints on the embedding is equivalent to constraining the forecasts? This should be clearly explained.
- The motivation for NMF should be clearly stated in the paper.
- More details on the baselines should be given, at least in the appendix.
- What does minibatch means with hierarchical time series data? This should be clearly explained.
- Given the simplified assumption, I am not sure the theoretical analysis really gives additional insight. Also, it only takes one paragraph in the whole paper.
- While simplified assumption are often required for a theoretical analysis, these assumptions shoul be clearly stated.
- "A small beta implies that the children time-series have structural similarities which is common in hierarchical datasets." -> I am not convinced by this sentence. Please give more details on Beta.
- Is Lemma 1 really useful? Replacing values in the Theorem should be enough.
- Simulation experiments could be used to confirm the usefulness of the method when indeed the data is low-dimensional. This would complement the theoretical analysis.


**Summary Of The Paper:**

The paper considers point forecasting of hierarchical time series, i.e. multivariate time series with hierarchical aggregation constraints. The authors propose a new approach based on decomposing the series along a global set of basis time series where (approximate) hierarchical constraints are applied on the coefficients of the basis decomposition. Forecasts are produced using a dynamic linear autoregressive model. Compared to existing state-of-the-art hierarchical models, the proposed approach improved overall performance on forecasts at different levels of the hierarchy on several public datasets.


**Summary Of The Review:**

While dimension reduction in hierarchical forecasting is an important problem, the paper lacks clarity and justifications for all the design choices. Major references are missing (dimension reduction for multivariate time series, probabilistic hierarchical forecasting, etc). The experimental setup is also questionable (loss function, etc). As a result, I do not think the paper is ready for publication.

---

> ### Author Response · Authors · 2021-11-15
> **Part 1/2: Thanks for the feedback**
>
> We thank the reviewer for the helpful comments and suggestions, this will help us improve the quality of our paper further. We address the comments roughly in order:
>
> - Coherent/Incoherent forecasts: Note that the coherence  property is not just a problem constraint, it is also considered to model the inductive biases in the hierarchical data which can lead to improved overall forecasting accuracy. This has been observed and noted in several previous papers (Wickramasuriya et al., 2019, Rangapuram et al., 2021). Hence we believe it is fair to compare coherent, approximately-coherent and incoherent methods on accuracy metrics.
> - Probabilistic prediction: We do agree that probabilistic prediction is harder and also very important, but that is outside the scope of this work. We would like to note here that similar to many prior works, in this paper we solely focus on point forecasts, and leave the extension to probabilistic forecasting to future work.
> - Basis decomposition on leaf time series: Modelling only the leaf time series using a basis decomposition and using bottom-up aggregation is indeed a valid option and will lead to perfectly coherent predictions. However, bottom up aggregation using the leaf level predictions does not always lead to a good accuracy on the higher levels, since in practical datasets, the leaf level time series can be quite noisy which results in a comparatively large error at higher levels due to accumulation of noise. As a result, jointly optimizing the model for all levels leads to better performance at the higher levels. We perform an additional experiment, where we do bottom up aggregation using the leaf level predictions, the results of which are shown below. We observe that bottom up aggregation leads to higher losses at the higher levels.
>
> | Model | Level 0 | Level 1 | Level 2 | Level 3 | Mean |
> |---------|----------|---------|---------|-----------|--------|
> | HiReD      | 0.04894/0.04881 | 0.05508/0.05300 | 0.07215/0.07752 | 0.27975/0.51184 | 0.11398/0.17279 |
> | HiReD + BU | 0.05835/0.05917 | 0.06790/0.07249 | 0.08388/0.09821 | 0.27975/0.51184 | 0.12247/0.18543 |
>
> We will add this bottom-up experiment as part of the ablation study in the final version of the paper.
> - Choice of loss function: In all our reported experiments, we do modify the training loss functions of all the baseline implementations to use the same loss function (MAE) for a fair comparison.
>
> The rest of the concerns are addressed in the next comment due to the character limit.

---

> ### Author Response · Authors · 2021-11-15
> **Part 2/2: Addressing rest of the comments**
>
> Continuation of the first response.
>
> Other comments:
> - Related work: We have included all related work on probabilistic hierarchical models known to us in sections 1 and 2. However, if you point us to the missing citations, we would be happy to include them in the discussion.
> - Figure 4: Tourism has 5 hierarchical levels, so L3 is the penultimate level to the leaf level. We have added this to the caption in the updated draft.
> - By asymptotically we mean asymptotic computational complexity (https://en.wikipedia.org/wiki/Asymptotic_computational_complexity). However, we sense that the theory section may not be clear. We have updated the discussion on the theory.
> - Insights from the theoretical analysis: Our theoretical results provide intuition about the utility of hierarchical regularization in our BD model. Despite the simplifying assumptions in Section 5 restricting the problem to the linear setting, our analysis (in Appendix A) characterizing the accuracy gains due to hierarchical regularization is non-trivial. Providing an improved theoretical analysis without our simplifying assumptions is highly non-trivial and beyond the scope of the current paper. However, we have now added more discussion about the theory in the updated draft of the paper (Appendix A.5).
> - About $\beta$: The $\beta$ assumption essentially bounds how far the embeddings of the leaf time series belonging to the same parent can be from each other. This is directly related to bounding how far the parent embedding is from the leaf embedding, as it is essentially the mean of the leaf nodes (mean property). In many practical scenarios, the children time series of a parent may not have too different seasonal trends (for example power consumption of houses in the same neighborhood, or sales of items under the same category) resulting in the parent time series following similar trends as well. We have added a further discussion about this in the paper (Appendix A.5).
> - Constraints on embeddings vs forecasts: The forecasts from the BD part of the model are linear in $\theta_i$. As a result, the fact that the parent embedding is the mean of the children embedding, leads to the parent forecasts being equal to the mean of the children forecasts. We have added a sentence to clarify this in the paper.
> - Non-negative matrix factorization: In many cases, the evolving global state of the set of time series may not be captured by the global covariates ($\mathbf{X}$) only. For instance, when there is an overall increase/decrease in sales across all time series, it is captured in the past values $\mathbf{Y}$ rather than $\mathbf{X}$. As a result, it may be required to feed in past values of time series to the BD model. However, since we cannot feed in the whole set of time series (order of 1000s) without leading to scalability issues, we choose some representative time series using NMF, thus approximating the global state. Our model is not limited by this choice though. One can also use more sophisticated method such as Temporal Latent Auto-Encoders (Nguyen et. al. 2021). We leave this idea for future exploration. We have added a paragraph on this in Appendix B.4.
> - Lemma 1: We do agree that Lemma 1 follows from replacing values in the Theorem, and we also clearly mention it in the proof and the main text. The main reason for labelling it as a Lemma is because we want to be able to refer to it in the main text without providing all the calculations.
> - Minor updates: We have added details about mini-batching in Appendix B.3. Thanks for pointing out the incorrect citation to Thomson et. al. for hierarchical coherence. We have updated the draft with a correct citation. We have split the related works section into two paragraphs now. We will add more details about the baselines in the appendix.
>
> We hope that our response satisfactorily addresses your concerns. Please let us know if you have any further comments.

---

> ### Comment · Reviewer_iRbQ · 2021-11-18
> **Response**
>
> We thank the authors for their response. Some additional comments are given below.
>
>
> > Coherent/Incoherent forecasts.
>
> I agree with the authors that coherency is often used as inductive bias. However, how do you choose between a method that has MSE=120 with incoherent forecasts and MSE=121 with coherent forecasts? The fact that optimal forecasts (conditional mean) are coherent suggest that coherency is a fundamental property of hierarchical forecasts. If you are only interested in accuracy for each series, then the hierarchy might not be the "best" inductive bias. Anyway, you should at least mention in your paper that you are only interested in forecast accuracy.
>
>
> > Basis decomposition on leaf time series.
>
> I was not asking for bottom-up predictions. My comment was related to basis decomposition of aggregated data.  To give a simple example, what would happen if you run PCA on X, Y and X+Y compared to PCA on X and Y, only? X+Y is clearly redundant.
>
>
> > Choice of the loss function. In all our reported experiments, we do modify the training loss functions of all the baseline implementations to use the same loss function (MAE) for a fair comparison.
>
> Was it possible to adapt all the methods: HIRED, RNN, DF, DeepGLO, DCRNN, RNN+ERM, L2 Emb, Hier-E2E? Also, I did not find the baseline implementations in your supplementary material.
>
> > Related work: We have included all related work on probabilistic hierarchical models known to us in sections 1 and 2. However, if you point us to the missing citations, we would be happy to include them in the discussion.
>
> Some state-of-the-art papers on hierarchical probabilistic forecasting are missing. For example, see the background section in Rangapuram et al., 2021.
>
> Panagiotelis et al. (2020)
> (Ben Taieb et al., 2017)
> ...
>
> Also, your paper is about basis decomposition in multivariate time series forecasting. You should cite related papers too (not just hierarchical). This is important since you do not produce coherent forecasts.

---

> ### Author Response · Authors · 2021-11-22
> **Addressing response 2**
>
> Thanks for the further comments.
> - Coherent/Incoherent forecasts: We have now further emphasized in the paper that forecast accuracy across all levels of the hierarchy is the main goal of this paper (end of page 2). We have also separated coherent and incoherent forecasting methods in the results table. We would also like to note here that coherent forecasting methods constitute a large section of prior work in hierarchical forecasting, and hence it is imperative to compare with those methods.
> - PCA: Your statement is indeed true in the case where the data perfectly lies in a low dimensional subspace. In practice, since the data is not perfectly low-rank, PCA on X and Y will not give the same results as PCA on X, Y and X+Y. As a result the basis learnt using only the leaf nodes is not exactly the same as the one learnt using all the nodes. In particular, we prefer to learn the basis using all the nodes since we simultaneously want  low reconstruction error on all time-series, not just the leaves. This choice is also more conducive from a practical implementation perspective: to jointly train the TVAR and Basis-Decomposition components, we use mini-batched gradient descent comprising both leaf and non-leaf time series in each batch. Training the Basis-Decomposition component separately on only the leaf nodes but TVAR on all nodes would add more implementation complexity around filtering non-leaf nodes.
> - Loss function: Our proposed model HiReD uses MAE, and DCRNN uses MAE by default. For the rest of the baselines, we do modify the loss function of ALL the baselines to use Mean Absolute Error as the training loss. We apologize for not explicitly specifying this in the doc earlier. We have now clarified this in Section 6 (para Baselines). We have also added a section in the appendix about the baselines.
> - Related work: Thanks for the references. We have now added the following citations:
>     - Hierarchical Forecasting: Taieb et al. 2017, Van Erven & Cugliari 2015, Panagiotelis et al. 2020
>     - Dimensionality reduction and multivariate time series: Salinas et al. 2019, Rasul et al. 2020, de Bezenac et al., 2020
>     - Others: Benidis et al. 2020
>
> Please let us know if any major issues remain. We look forward to addressing all your major concerns and hope that you consider updating your score. Thank you once again

---

### Official Review · Reviewer_U9mT · 2021-11-03

**Correctness:** 3
**Technical Novelty And Significance:** 2
**Empirical Novelty And Significance:** 3
**Recommendation:** 6
**Confidence:** 4

**Main Review:**


- The assumption $|| \theta_n - \theta_0 ||_2 \leq \beta$ seems to be restrictive. The embedding parameters could satisfy the hierarchical structure while the parent embedding parameter $\theta_0$ would be different that the children embedding $\theta_n$. Authors mention at the top of page 7 that as $\beta \rightarrow 0$, the hierarchically regularized estimator approaches an error $L$ times smaller when compared to the un-regularized estimator. That seems to be more related to the assumption than the developed algorithm. For very small $\beta$, essentially, the sample size is increased from $T$ to $LT$ since $L$ additional time series are observed, but thanks to the assumption that $\beta$ is very small, the number of parameters to be estimated did not increase. As a result, the consistency rate for estimating the embedding parameter will be of order $O(1/(LT))$ instead of $O(1/(T))$. Thus, the theoretical justification of the proposed methodology is unclear.

- Results of Theorem 1 are related to prediction error consistency rather than parameter estimation consistency. It is appropriate to consider verifying the rate for $|| \tilde{\theta}_n - \theta_n ||_2$ instead of  $|| \tilde{\theta}_n - \theta_n ||_\Sigma$.

- It is not clear why the Non-Negative Matrix Factorization (NMF) algorithm is used to select a small set
of representative time series that encode the global state, please elaborate further.

- Typo in page 7: "Tables 1 and 2 shows ..."

- Typo in page 9: "... state of the art baselines for most levels of the hierarchically."

**Summary Of The Paper:**

In this manuscript, the authors proposed a method for hierarchical time series forecasting, consisting of two
components, the TVAR model, and the BD model. Further, the performance of the proposed algorithm is empirically evaluated on three benchmark datasets and showed that the proposed model consistently improved over state of the art baselines.

**Summary Of The Review:**


- The paper is well written and easy to read.
- Theoretical results need clarification, while some assumptions need to be relaxed.
- Code for reproducing the results is provided together with links to publicly available real data sets used in the manuscript.

---

> ### Author Response · Authors · 2021-11-15
> **Thanks for the feedback**
>
> Thank you for the detailed comments, suggestions and the positive feedback. We address all your comments below.
>
> - About $\beta$: The $\beta$ assumption essentially bounds how far the embeddings of the leaf time series belonging to the same parent can be from each other. This is directly related to bounding how far the parent embedding is from the leaf embedding, as it is essentially the mean of the leaf nodes (mean property). In many practical scenarios, the children time series of a parent may not have too different seasonal trends (for example power consumption of houses in the same neighborhood, or sales of items under the same category) resulting in the parent time series following similar trends as well.
> Note that we stated the limiting case of $\beta \to 0$ only for the purpose of intuition, which exactly captures the argument you mentioned. However our result shows that as long as $\beta = o(\sqrt{K/T})$, we obtain an improvement over the non-hierarchical error bound of $O(K/T)$. We believe this statement is not as obvious. We have expanded this section further in the paper (Appendix A.5).
> - Prediction error vs estimation error: We are indeed more interested in the prediction error rather than parameter estimation error, since the former is what quantifies the performance of the model in practice. This is quite standard in analysis of linear regression as well (Martin Wainwright book includes further details). Moreover, the prediction error can also be used to bound the estimation error as $\\|\widehat{\theta} - \theta\\|_\Sigma^2 \ge \lambda\_{\min}(\Sigma) \\|\hat{\theta} - \theta\\|_2^2$. However, this requires the assumption that $\lambda\_{\min}(\Sigma) > 0$, since otherwise the parameters cannot be identified anyway due to degeneracy. We dave also added a discussion about this in Appendix A.5.
> - Non-negative matrix factorization: In many cases, the evolving global state of the set of time series may not always be captured by the global covariates ($\mathbf{X}$) only. For instance, when there is an overall increase/decrease in sales across all time series, it is captured in the past values $\mathbf{Y}$ rather than $\mathbf{X}$. As a result, it may be required to feed in past values of time series to the BD model. However, since we cannot feed in the whole set of time series (order of 1000s) without leading to scalability issues, we choose some representative time series using NMF, thus approximating the global state. Our model is not limited by this choice though. One can also use more sophisticated methods such as Temporal Latent Auto-Encoders (Nguyen et. al. 2021). We leave this idea for future exploration. We have added a discussion about this in Appendix B.4.
> - Thanks for pointing out the typos, we have updated the paper with the corrections.
>
> Please let us know if there are any more concerns

---

### Author Response · Authors · 2021-11-25
**Summary of updates**

We thank the reviewers for their valuable comments on the paper. We appreciate that the reviewers found our paper “well written and easy to read” (U9mT), “conveys a good reader journey experience” (UnFM), solves “an important problem” (iRbQ), “scales well” (UnFM), “easy to implement” (UnFM), and “showing good results against relevant baselines” (UnFM).

We have now updated the paper to address the reviewers’ questions. In particular,
- We have added an 8th baseline SHARQ (Han et. al. 2021) in Table 1.
- We have also rewritten the discussion on Theorem 1 in Section 5 to address the raised questions. In particular, we would like to re-emphasize that we study an idealized linear setting which nevertheless highlights some key insights for our basis decomposition. Despite the simplifying assumptions, our analysis (in Appendix A) characterizing the accuracy gains due to hierarchical regularization, is non-trivial.
- We note that while probabilistic forecasting is an interesting problem, in this paper we specifically focus on point forecasts only. We also emphasize that our primary goal is to improve prediction accuracy across all levels. Using the coherency constraints as an inductive bias helps in that regard.

Other clarifications and details added to the paper:
- Further details about using NMF (Appendix B.4).
- Further details about training and baselines have been added to Appendix B.3 and B.5 to complement the code already submitted in the supplementary material.
- Missing references related to hierarchical forecasting and dimensionality reduction were added.

---

### Decision · Program_Chairs · 2022-01-20

**Decision:**

Reject

**Comment:**

The paper proposes a method for time series forecasting based on a hierarchical deep learning approach. Three reviewers submitted reviews, with two marginally accept and one marginally reject. The paper was therefore borderline, but the issues raised by the marginal reject reviewer on the justification for the design choice of a deep latent model and the experimental setup appear worth addressing in a revision resubmitted to another conference.